# Systematic analysis of paralogous regions in 41,755 exomes uncovers clinically relevant variation

Wouter Steyaert[1,2], Lonneke Haer-Wigman[1], Rolph Pfundt[1], Debby Hellebrekers[3], Marloes Steehouwer[1], Juliet Hampstead[1], Elke de Boer [1,4], Alexander Stegmann [3], Helger Yntema [1], Erik-Jan Kamsteeg [1], Han Brunner[1,3], Alexander Hoischen [1,2,5,6] & Christian Gilissen [1,2,6] ✉

The short lengths of short-read sequencing reads challenge the analysis of paralogous genomic regions in exome and genome sequencing data. Most genetic variants within these homologous regions therefore remain unidentified in standard analyses. Here, we present a method (Chameleolyser) that accurately identifies single nucleotide variants and small insertions/deletions (SNVs/Indels), copy number variants and ectopic gene conversion events in duplicated genomic regions using whole-exome sequencing data. Application to a cohort of 41,755 exome samples yields 20,432 rare homozygous deletions and 2,529,791 rare SNVs/Indels, of which we show that 338,084 are due to gene conversion events. None of the SNVs/Indels are detectable using regular analysis techniques. Validation by high-fidelity long-read sequencing in 20 samples confirms >88% of called variants. Focusing on variation in known disease genes leads to a direct molecular diagnosis in 25 previously undiagnosed patients. Our method can readily be applied to existing exome data.

Over 1700 human protein-coding genes partly or completely share a very high sequence identity with other genomic regions[1]. These paralogous regions originate from small- or large-scale duplication events or retro-transpositions in the evolution of the human species. The sequence and function of these duplicated genomic regions typically diverge over evolutionary time by the accumulation of mutations at different rates. One of the copies might lose its function and evolve to a non-coding paralog (a pseudogene) or to a coding paralog with a different function[2,3].

A well-known genetic mechanism that is relevant when studying paralogous regions is ectopic gene conversion. A non-allelic or ectopic gene conversion is an event where a sequence is copied from a specific genomic region (the donor region) to a distant region (the acceptor region). When the donor and acceptor sequence differ, this introduces new genetic variation into the acceptor site[4,5]. Ectopic gene conversions occur in at least 1% of human genes associated with inherited disease[6]. In several of these genes such as *STRC*, *OTOA*, and *SMN1* gene conversions have previously been identified as a cause of genetic disease[7–9].

Despite their clinical relevance, gene conversions remain unidentified in the analysis of short-read data such as whole-exome sequencing (WES) and whole-genome sequencing (WGS) data. Indeed, in case of an ectopic gene conversion, the sequencing reads that originate from the acceptor site will align onto the reference sequence

[1]Department of Human Genetics, Radboud Institute for Molecular Life Sciences, Radboud University Medical Center, Geert Grooteplein 10, 6525 GA Nijmegen, The Netherlands. [2]Radboud Institute for Molecular Life Sciences, Nijmegen, Netherlands. [3]Maastricht University Medical Center + , Department of Clinical Genetics, Maastricht, Netherlands. [4]Radboud University, Donders Institute for Brain, Cognition and Behaviour, Nijmegen, Netherlands. [5]Radboud University Medical Center, Department of Internal Medicine and Radboud Center for Infectious Diseases (RCI), Nijmegen, Netherlands. [6]These authors jointly supervised this work: Alexander Hoischen, Christian Gilissen. ✉e-mail: christian.gilissen@radboudumc.nl

corresponding to the donor site. As a result, no reads will be aligned to the acceptor site and single nucleotide variants and small insertions and deletions (SNVs/Indels) that are introduced by means of the gene conversion remain unidentified (Fig. 1e). Copy number variant (CNV) callers, however, will typically identify such events as deletions despite the fact that no deletion is present in the patient's DNA (from here the term 'deletion' refers to genetic events with the size of single or multiple exons).

The issue of variant discovery within paralogous regions is not limited to gene conversions. SNVs/Indels that are not introduced by means of a gene conversion also remain undetected, especially in genomic regions that have an identical paralog (100% sequence identity). In such cases, short sequencing reads align equally well to multiple locations in the genome and will typically be assigned a mapping quality of zero. These reads will be ignored by the variant calling algorithm as their alignment is deemed ambiguous. As a result, genetic variants that are supported by these reads are not detected (Fig. 1b).

Among the methods that enable the identification of CNVs in WES and WGS data, a limited number is specifically designed to estimate copy numbers of paralogous genes[10,11]. By using the read depth at singly unique nucleotides (SUNs; the sequence differences between paralogs), it is possible to genotype the copy and content of paralogs within duplicated gene families. For regions that have an identical copy elsewhere in the genome (without SUNs), an estimate of the total copy number can be made. Despite these methods being accurate, they are designed to run on WGS data and they do not explicitly identify gene conversions. Furthermore, to our knowledge, there are currently no methods available to identify SNVs/Indels in identical paralogs. Ebbert et al., 2019 performed a thorough characterisation of paralogous regions in the human genome and suggested a strategy for rescuing variants in these regions based on re-alignment of reads to a masked reference genome, but their work did not provide a concrete solution[12]. For these reasons, the accurate sequence analysis of paralogous regions still relies on experimental assays that only include one or a couple of genes. Typically, specific polymerase chain reaction

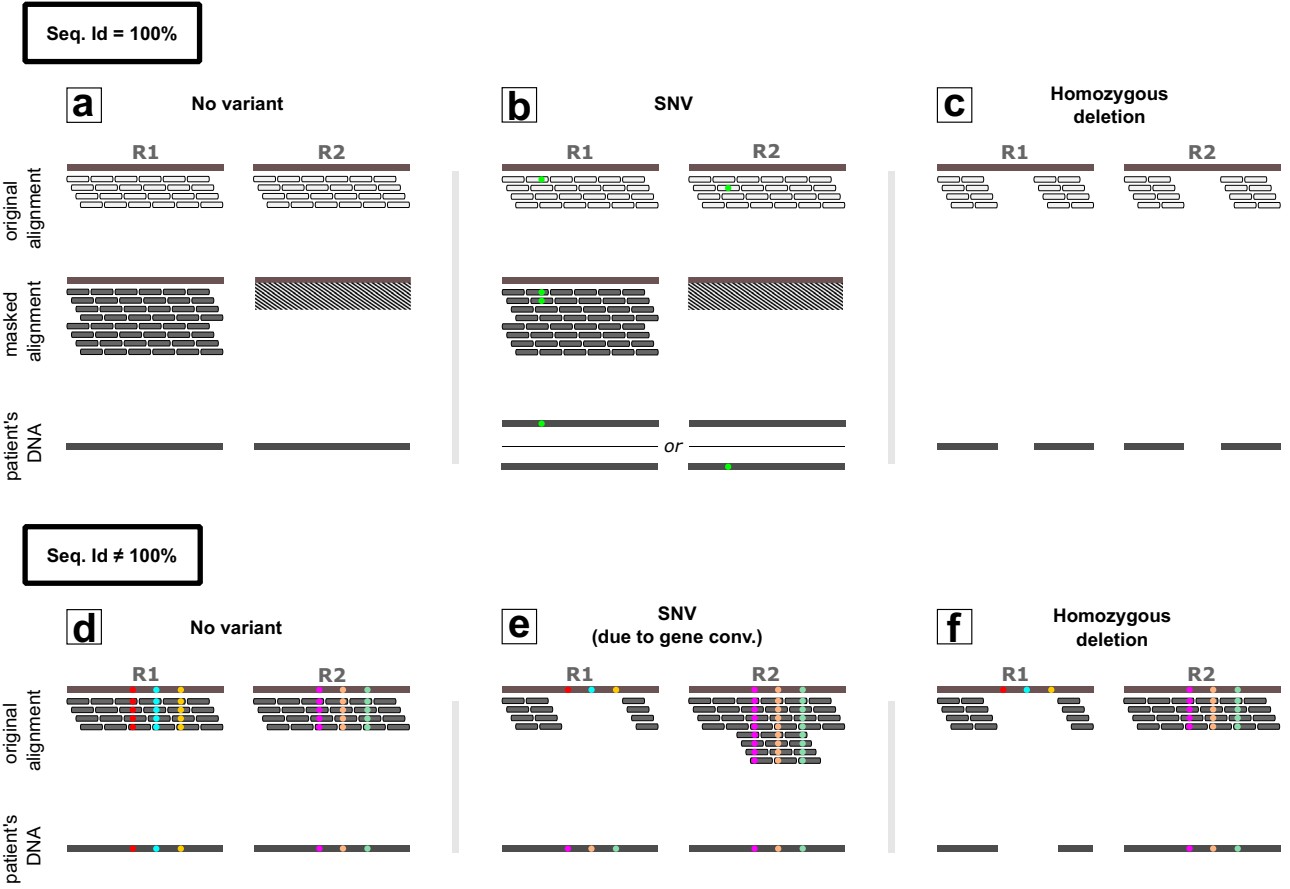

**Fig. 1 | Schematic overview of genetic events that are identified by Chameleolyser.** Regions R1 and R2 are two regions with a very high sequence identity. In panels **a**, **b** and **c** these two regions are completely identical (Seq. Id = 100%). As a consequence, reads that align onto these regions will have mapping qualities of 0 (when no masking is applied). To indicate this, reads are displayed white. Within Chameleolyser, reads are extracted and re-aligned onto a reference sequence in which R2 is masked. As a result, reads align uniquely onto R1 and will have mapping scores different from 0. This is indicated by representing them in grey. By applying a sensitive variant calling onto this masked alignment, Chameleolyser is able to identify single nucleotide variants and small indels (SNVs/Indels; green bullet in panel **b**). Nevertheless, the exact position of the variant remains ambiguous, hence we named them VAPs (variant with ambiguous position). In case R1 and R2 are identical in sequence, Chameleolyser limits the identification of homozygous deletions to events in which both R1 and R2 are deleted (panel **c**).

Panels **d**, **e** and **f** illustrate the scenarios in which R1 and R2 are not completely identical (Seq. Id ≠ 100%). The three positions in which R1 differs from R2 are indicated with a coloured bullet. Since reads that align onto these regions will have sufficiently good mapping qualities, the identification of regular SNVs/Indels does not pose a problem for standard data analysis pipelines. Nevertheless, SNVs/Indels that result from a gene conversion typically remain unidentified. By only considering the coverage profile of R1, an ectopic gene conversion and a deletion look identical (panels **e** and **f**). Chameleolyser also considers the coverage at locus R2. As a result, gene conversions can be distinguished from deletions. Indeed, in case of an ectopic gene conversion, reads that originate from the acceptor site will align onto the reference sequence of the donor site resulting in an increased sequencing coverage as opposed to the scenario where no gene conversion is present.

(PCR) primers are designed to generate long-range PCR fragments that span the paralogous regions. Despite the fact that this approach has been successful for quite a number of genes these assays remain challenging to design and laborious to perform and are therefore not applied at scale[1,13,14].

Here, we present a method (Chameleolyser) that enables the identification of SNVs/Indels, CNVs, and ectopic gene conversions in all paralogous regions in the coding portions of the human genome based on short-read sequencing data. By applying Chameleolyser to a cohort of 41,755 WES samples, we identify an average of 60 genetic variants per sample that could not be detected using standard WES analysis. Validation by high-fidelity long-read sequencing in 20 samples confirms >88% of called variants. Stringent filtering and clinical interpretation of these variants results in a genetic diagnosis for 25 previously undiagnosed rare disease patients. The wider application of our method might result in a new reservoir of genetic variation from which new biological insights could be gained. Chameleolyser is implemented in Perl5 and requires a BAM or CRAM file (relative to GRCh37) as input. It runs about one hour on a single core for a single sample (depending on the enrichment kit and sequencing depth). Both raw and filtered variants are written to a tab-separated file. The tool is freely available on GitHub (https://github.com/Genome-Bioinformatics-RadboudUMC/Chameleolyser) where also installation and usage instructions can be found[15].

## Results

Chameleolyser works by extracting reads in the 3.5% of the exome that is affected by sequence homology (paralogous regions (Methods)) and re-aligning them to a reference genome in which all but one paralogs within each set of paralogs are masked[12]. By masking all nucleotides in these regions in the reference genome, no sequencing reads will be aligned onto them. As a result, all reads that originate from a set of paralogous sequences are uniquely aligned onto a single region in the reference genome (the non-masked region; Fig. 1b). Subsequently we perform sensitive variant calling to identify SNVs/Indels (Methods).

Homozygous deletions and ectopic gene conversion events are identified by analysing the coverage profile in the original alignment (without masking). In short-read sequencing data, a homozygous deletion and the acceptor site of a homozygous ectopic gene conversion appear identical: no reads are aligned onto that site of the reference genome. By also considering the number of reads that align onto the paralogous regions, it is possible to discriminate between deletions and gene conversions. In case of ectopic gene conversion, the reads that originate from the acceptor site align onto the reference sequence of the donor site which results in a twofold increase in sequencing depth relative to what is expected (Fig. 1e, f). By applying this approach to a dataset of 41,755 exome samples we identified 2,191,707 SNVs/Indels which are not due to a gene conversion (cohort allele frequency (CAF) ≤ 10%; Supplementary Fig. 1, Table 1, Supplementary Data 1), 22,600 homozygous gene conversions that jointly

**Table 1 | Observed number of variant calls, VAPs and deletions with two different cohort allele frequency thresholds (10% and 0.5%) in our cohort of 41,755 exome samples**

| Variant type | Due to ectopic gene conversion | Within or outside an OMIM disease gene | Transcript consequence | VAF threshold | Number of variants/VAPs |
|---|---|---|---|---|---|
| deletion | NA | Within | NA | 0.10 | 6250 |
| deletion | NA | Outside | NA | 0.10 | 14,182 |
| SNV/Indel | Yes | Within | LoF | 0.10 | 1043 |
| SNV/Indel | Yes | Within | Miss | 0.10 | 4507 |
| SNV/Indel | Yes | Within | Rest | 0.10 | 56,279 |
| SNV/Indel | Yes | Outside | LoF | 0.10 | 341 |
| SNV/Indel | Yes | Outside | Miss | 0.10 | 10,970 |
| SNV/Indel | Yes | Outside | Rest | 0.10 | 264,944 |
| SNV/Indel | No | Within | LoF | 0.10 | 13,875 |
| SNV/Indel | No | Within | Miss | 0.10 | 142,324 |
| SNV/Indel | No | Within | Rest | 0.10 | 524,376 |
| SNV/Indel | No | Outside | LoF | 0.10 | 53,908 |
| SNV/Indel | No | Outside | Miss | 0.10 | 514,659 |
| SNV/Indel | No | Outside | Rest | 0.10 | 3,347,319 |
| deletion | NA | Within | NA | 0.005 | 1182 |
| deletion | NA | Outside | NA | 0.005 | 3885 |
| SNV/Indel | Yes | Within | LoF | 0.005 | 181 |
| SNV/Indel | Yes | Within | Miss | 0.005 | 1279 |
| SNV/Indel | Yes | Within | Rest | 0.005 | 22,831 |
| SNV/Indel | Yes | Outside | LoF | 0.005 | 341 |
| SNV/Indel | Yes | Outside | Miss | 0.005 | 480 |
| SNV/Indel | Yes | Outside | Rest | 0.005 | 60,298 |
| SNV/Indel | No | Within | LoF | 0.005 | 2000 |
| SNV/Indel | No | Within | Miss | 0.005 | 20,827 |
| SNV/Indel | No | Within | Rest | 0.005 | 79,037 |
| SNV/Indel | No | Outside | LoF | 0.005 | 8760 |
| SNV/Indel | No | Outside | Miss | 0.005 | 79,962 |
| SNV/Indel | No | Outside | Rest | 0.005 | 482,720 |

All variants were annotated on Ensembl canonical transcripts. Loss-of-function (LoF) and missense (Miss) variants are relative to these transcripts.

introduce an additional 338,084 SNVs/Indels (CAF ≤ 10%; Supplementary Fig. 2, Table 1, Supplementary Data 2 and 3) and 20,432 homozygous copy number losses (CAF ≤ 10%; Supplementary Fig. 3, Table 1, Supplementary Data 4). Importantly, none of the SNVs/Indels, either being the result of a gene conversion or not, were detected by a standard WES analysis (Methods).

## Validation

To technically validate our variant call set, we performed whole genome high-coverage long-read sequencing (LRS) for 20 samples using PacBio high-fidelity technology[16]. Within this subset of samples, Chameleolyser identified 769 SNV/Indel calls that are not the result of a gene conversion. LRS data confirmed 678 of these calls (88.2%; Fig. 2, Methods, Supplementary Data 5). Of the 120/769 rare SNVs/Indels (CAF ≤ 0.5%), 111 (92.5%) are concordant with the LRS data (Supplementary Data 5). Our analysis furthermore identified 8 homozygous gene conversions and 15 homozygous deletions within the subset of samples for which LRS data was generated. LRS data confirmed all ectopic gene conversions (100%) and 13/15 homozygous deletions (86.7%) (Fig. 2, Supplementary Data 6, Methods).

The quality of our variant call set was further evaluated by using the 6980 parent-offspring trios that are present in our dataset. We observe that 99.0% of the SNVs/Indels that are present in the offspring is also called in one of the parents (Methods, Supplementary Data 7). This suggests that only a small fraction of our variant calls are technical artifacts.

In addition to our in-house validation samples we also applied Chameleolyser to 5 genome-in-a-bottle samples (Methods). Since the identification of deletions and gene conversions requires a larger number of samples enriched with the same enrichment kit, the precision analysis was restricted to SNVs/Indels (not the result of a gene conversion). From the 118 SNV/Indel calls made by Chameleolyser, 98 are concordant with LRS (83.1%; Methods, Supplementary Data 8). From the 39 calls corresponding to rare SNVs/Indels, 35 were concordant with LRS (89.7%; Supplementary Data 8).

## Comparison with other variant callers

Chameleolyser's ability to identify SNVs/Indels (not the result of a gene conversion) was compared with GATK and DeepVariant[17]. The sensitivity for both of these tools is exactly zero within genomic regions that are associated with zero mapping qualities in WES (Supplementary Data 9, Fig. 1b). With Chameleolyser a sensitivity of 43% is achieved (Methods). In regions onto which sequencing reads align uniquely, it has been shown that GATK and DeepVariant are excellent tools for the identification of SNVs/Indels[18]. Within these regions, the added value of Chameleolyser is limited with a sensitivity of 88.0% compared to 86.3% for GATK (Methods).

Sensitivity could not be assessed for homozygous deletions and ectopic gene conversions since we cannot, due to the availability of only a limited number of long-read sequencing samples, derive a call set of high-quality events with a population allele frequency ≤0.10 (Methods). The unique value of Chameleolyser can however be demonstrated by comparing its output with ExomeDepth[19] and Conifer[20] (Methods, Supplementary Fig. 4). Within the 20 in-house samples for which LRS alignments were generated, there are 4 events (3 deletions and 1 gene conversion) that are only called by Chameleolyser. Of these, 2 events (1 deletion and 1 conversion) were concordant with the LRS alignments. The other 12 deletions and 7 conversions that were identified by Chameleolyser are all called as deletions by ExomeDepth. As opposed to Conifer (for which there are no homozygous deletion calls within the validation samples), ExomeDepth made an additional 201 homozygous deletion calls which were not made by the other tools. Based on the LRS alignments we estimated the precision at 32.5% (Methods, Supplementary Data 10).

## Variants with ambiguous positions

Heterozygous SNV/Indel calls (not due to a gene conversion and not corresponding to SUNs (methods)) result from a genomic alteration in one of the paralogs within the respective set of paralogs (Fig. 1b). Since short-read data does not contain the information to discriminate between the different paralogs in an identical set of paralogs, all possible variants that could have caused the variant call are computed and annotated (Methods). In the remainder of the text we will call these "variants with ambiguous positions" (VAPs). This uncertainty is not applicable for variants which are homozygous in all paralogs nor is it relevant for gene conversions and deletions since these events are identified based on coverage data (Methods).

Approximately 10% of VAPs originate from protein-altering variants or from the corresponding alteration in only one possible non-coding paralog (the variant thus resides in a set of 2 paralogs of which one is coding and the other is not). In principle we would expect that half of these VAPs actually reside in the coding region. However, selection may act more on coding regions which could lead to an overrepresentation of VAPs that are actually present in non-coding regions. In order to derive the fraction of VAPs that originate from protein-altering variants we used two different approaches. Firstly, by using the LRS data that we used for validation purposes we can determine the actual location of these VAPs and thus determine the fraction. Within the 20 WES samples for which we generated LRS data, we identified 65 VAPs satisfying the aforementioned criteria. From these, 25 (38%) turned out to be present in the coding regions (p-value$_{binom}$ = 0.08; Supplementary Data 11).

A second approach to estimate the fraction of coding variants among VAPs uses the ratios of synonymous, missense and loss-of-function (LoF) variants in the paralogous and non-duplicated (unique) regions of the exome. Using our standard WES analysis pipeline[21] we find exome-wide ratios of 1.19 and 0.043 for missense to synonymous and LoF to synonymous variants respectively (Supplementary Data 12). Within the homologous regions of the exome Chameleolyser identified on average 6.3 synonymous, 11.1 missense and 1.40 LoF VAPs per sample (Supplementary Data 13). We assume that synonymous variants are not under strong selection and thus that half of synonymous VAPs actually originate from variants residing in protein coding regions. If we further assume that the ratios of missense to synonymous and LoF to synonymous variants are comparable between paralogous and non-paralogous regions, we can calculate the proportion of missense and LoF variants among VAPs as 33.8% and 10.0%, or 3.75 and 0.14 variants per sample respectively. As such, we provide two lines of evidence that roughly 30-40%, of protein-altering VAPs resides in protein coding regions.

## Systematic analysis of SNVs/Indels results in 14 diagnoses

In order to investigate SNVs/Indels that could be of clinical interest we only considered variants in exomes of patients that were molecularly undiagnosed (n = 17,650; Supplementary Data 14). We selected missense and LoF VAPs with a CAF ≤ 0.5%, and occurring in clinically relevant genes according to predefined gene panels for which an investigation was requested for the particular patient. In addition, we included a single synonymous variant in *SMN1* (chr5:g.70,247,773 C > T(GRCh37)) that is known to lead to a truncated protein product[22].

The application of the aforementioned filter criteria to our variant call set resulted in 1071 heterozygous VAPs (131 LoF and 940 missense; Supplementary Data 15) as well as 57 homozygous variants (5 LoF, 46 missense and 6 synonymous; Supplementary Data 16). All of the homozygous variants are introduced in the gene of interest by means of gene conversions that most likely occurred in a proximal or distant ancestor (a total of 21). Importantly, the genomic positions of these homozygous variants are not ambiguous (hence these are not VAPs), but clear site-specific calls (Fig. 1d, e).

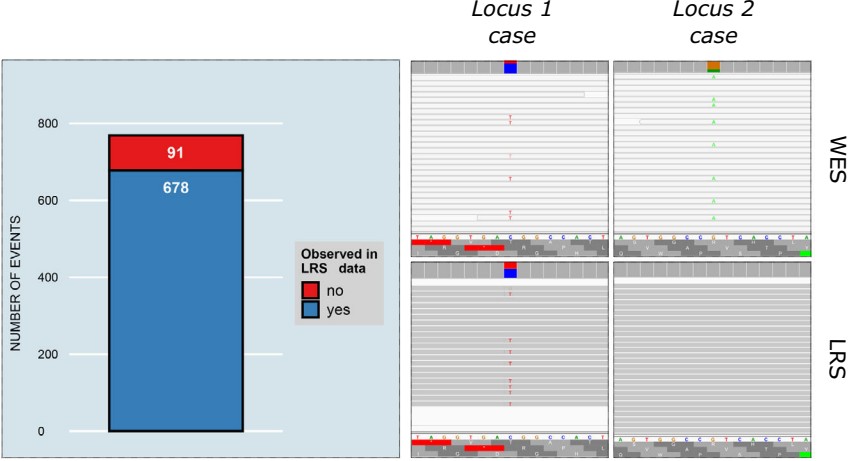

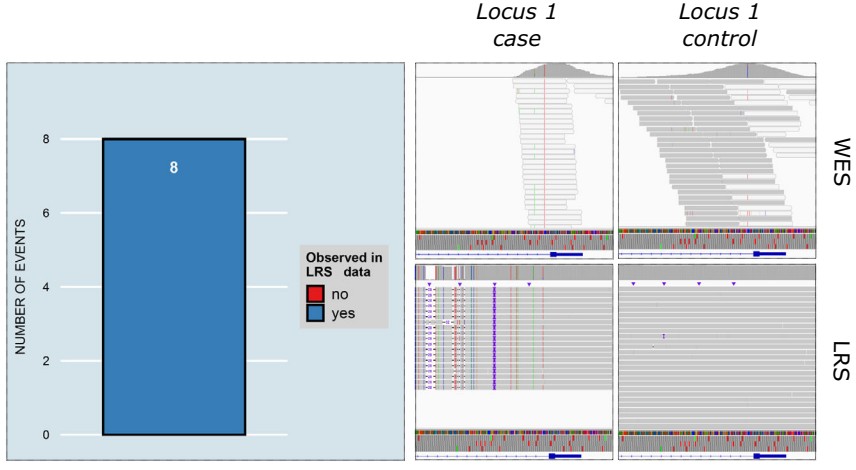

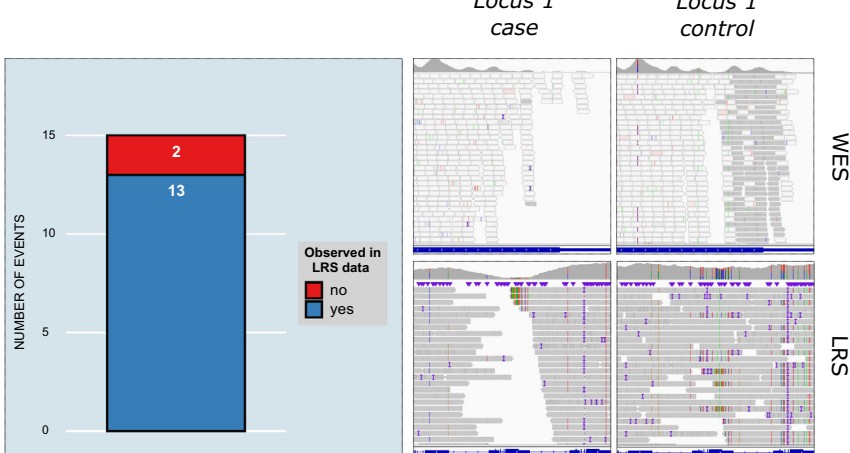

Among the 1071 rare VAPs that we identified in our cohort there were 7 alterations in the *STRC* gene that occur in patients in which we also identified a heterozygous multi-exonic deletion (Supplementary Data 17, Supplementary Fig. 5). Validation experiments consisting of multiplex ligation-dependent probe amplification (MLPA) and long-range polymerase chain reaction (PCR) followed by sequencing were conducted for all of the 7 individuals.

This confirmed that all of the 7 deletions and 4 out of the 7 SNVs/Indels (1 LoF, 3 missense−all in trans with the deletion) were present in the *STRC* gene (and thus not in its pseudogene; Table 2; Fig. 3; Supplementary Fig. 6), resulting in 4 genetic diagnoses. The other 1064 VAPs did not reveal any additional diagnosis. Either the phenotype that is associated with the gene of interest did not sufficiently match the clinical presentation of the patient or the

**Fig. 2 | Overview of validation successes with LRS.** Each variant type (single nucleotide variant and small indel (SNV/Indel) not due to a gene conversion, homozygous gene conversion and homozygous deletion) is accompanied with a bar chart and one concrete example. Within each bar chart, the correspondence between Chameleolyser and long-read sequencing (LRS) is shown. The genomic coordinates of the SNV (not due to a gene conversion) in the IGV screenshots is chr2:96,692,489-C/T (locus 1) and chr2:96,463,586-G/A (locus 2). In whole-exome-sequencing (WES) we observe that roughly 25% of the reads at each locus support the variant allele. Based on LRS we clearly see that the genomic alteration is present as a heterozygous SNV in locus 1 (and not in locus 2). The genomic coordinate of the homozygous gene conversion that is shown in the IGV screenshot is chr1:22,338,347-22,339,613 (*CELA3A*; locus specific). WES data shows a clear difference between a sample with (a case) and a sample without the event (a control). In the case (as opposed to the control) there is not any read that uniquely aligns onto the beginning of intron 1. Considering the LRS data, we see that this region is not deleted in the case sample. In contrast, we see several SNVs/Indels which are absent in the control sample. These alterations indeed correspond to sequence differences between *CELA3A* and *CELA3B* (Supplementary Data 22). A conversion from *CELA3B* to *CELA3A* is responsible for these SNVs/Indels being present in *CELA3A* in the case sample. The genomic coordinate of the homozygous deletion that is shown in the IGV screenshot is chr9:84,545,162-84,547,705. The difference between a case and a control sample can be seen in WES. This corresponds to the LRS data.

## Table 2 | Overview of new genetic diagnosis in our study cohort as a consequence of disease-causing variations identified with Chameleolyser

| Sample | Chrom | Start | End | TypeOfEvent | GeneSymbol | OMIM |
|---|---|---|---|---|---|---|
| SAMPLE_24323 | chr16 | 21747381 | 21747911 | Conversion | *OTOA* | Deafness, autosomal recessive 22 |
| SAMPLE_29813 | chr16 | 21747381 | 21747911 | Conversion | *OTOA* | Deafness, autosomal recessive 22 |
| SAMPLE_30025 | chr16 | 21747381 | 21747911 | Conversion | *OTOA* | Deafness, autosomal recessive 22 |
| SAMPLE_26907 | chr5 | 70247601 | 70248925 | Conversion | *SMN1* | Spinal muscular atrophy-1–4 |
| SAMPLE_28821 | chr5 | 70247601 | 70248925 | Conversion | *SMN1* | Spinal muscular atrophy-1–4 |
| SAMPLE_36286 | chr5 | 70247601 | 70248925 | Conversion | *SMN1* | Spinal muscular atrophy-1–4 |
| SAMPLE_37053 | chr5 | 70247601 | 70248925 | Conversion | *SMN1* | Spinal muscular atrophy-1–4 |
| SAMPLE_39455 | chr5 | 70247601 | 70248925 | Conversion | *SMN1* | Spinal muscular atrophy-1–4 |
| SAMPLE_20848 | chr5 | 70247601 | 70248925 | Conversion | *SMN1* | Spinal muscular atrophy-1–4 |
| SAMPLE_23606 | chr15 | 43890861 | 43897797 | Conversion | *STRC* | Deafness, autosomal recessive 16 |
| SAMPLE_37062 | chr16 | 21747381 | 21747911 | Deletion | *OTOA* | Deafness, autosomal recessive 22 |
| SAMPLE_37080 | chr16 | 21747381 | 21747911 | Deletion | *OTOA* | Deafness, autosomal recessive 22 |
| SAMPLE_23649 | chr5 | 70247601 | 70248925 | Deletion | *SMN1* | Spinal muscular atrophy-1–4 |
| SAMPLE_6943 | chr5 | 70247601 | 70248925 | Deletion | *SMN1* | Spinal muscular atrophy-1–4 |
| SAMPLE_27880 | chr5 | 70247601 | 70248925 | Deletion | *SMN1* | Spinal muscular atrophy-1–4 |
| SAMPLE_9901 | chr5 | 70247601 | 70248925 | Deletion | *SMN1* | Spinal muscular atrophy-1–4 |
| SAMPLE_29108 | chr5 | 70247601 | 70248925 | Deletion | *SMN1* | Spinal muscular atrophy-1–4 |
| SAMPLE_30394 | chr5 | 70247601 | 70248925 | Deletion | *SMN1* | Spinal muscular atrophy-1–4 |
| SAMPLE_31929 | chr5 | 70247601 | 70248925 | Deletion | *SMN1* | Spinal muscular atrophy-1–4 |
| SAMPLE_31987 | chr5 | 70247601 | 70248925 | Deletion | *SMN1* | Spinal muscular atrophy-1-4 |
| SAMPLE_40265 | chr5 | 70247601 | 70248925 | Deletion | *SMN1* | Spinal muscular atrophy-1–4 |
| SAMPLE_21563 | chr15 | 43908399 | 43908399 | hemizygous SNV/Indel: G > C | *STRC* | Deafness, autosomal recessive 16 |
|  |  | 43890861 | 43894856 | heterozygous deletion |  |  |
| SAMPLE_32502 | chr15 | 43906154 | 43906154 | hemizygous SNV/Indel: G > C | *STRC* | Deafness, autosomal recessive 16 |
|  |  | 43890861 | 43894856 | heterozygous deletion |  |  |
| SAMPLE_38648 | chr15 | 43908409 | 43908409 | hemizygous SNV/Indel: G > - | *STRC* | Deafness, autosomal recessive 16 |
|  |  | 43890861 | 43894856 | heterozygous deletion |  |  |
| SAMPLE_36262 | chr15 | 43908184 | 43908184 | hemizygous SNV/Indel: C > G | *STRC* | Deafness, autosomal recessive 16 |
|  |  | 43890861 | 43894856 | heterozygous deletion |  |  |

The first column indicates in which sample the variant was identified. The second, third and fourth column respectively represent the chromosome, genomic start and end of the event (hg19). In the next column, the type of genetic event can be found. The sixth column indicates the respective gene symbol and in the last column the associated disease is displayed.

disease gene is recessive where only a heterozygous variant is identified.

Out of the 21 ectopic gene conversions that were interpreted, 11 were considered as not causal for disease due to their frequency among the patients for which the specific gene was not a gene of interest. The other 10 conversions provided a direct diagnosis (Table 2; Fig. 3; Supplementary Fig. 7; Supplementary Fig. 8). All of these events were found in one of only three genes: *STRC* (*n* = 1), *OTOA* (n = 3) and *SMN1* (*n* = 6). The conversion from *STRCP1* to *STRC* causes a LoF variant to be introduced in *STRC* and thus leads to a null allele[7]. The 3 gene

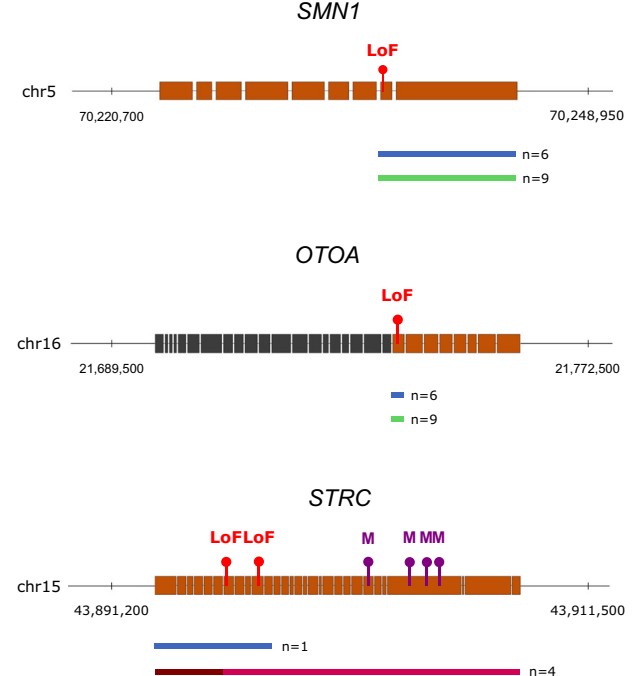

**Fig. 3 | Overview of previously unidentified disease-causing variants.** The three genes in which we identified disease-causing variants are represented by a model of their Ensembl canonical transcript. The orange parts of the gene are affected by sequence homology (thus incorporated in our analysis). The black parts are not. Homozygous gene conversions are illustrated with blue rectangles (*n* = 10). Loss-of-function variants (LoFs) which are introduced by means of these gene conversions are indicated with a red bullet. Homozygous deletions are indicated with green rectangles (*n* = 11). The bordeaux rectangles underneath *STRC* represent a heterozygous deletion (*n* = 4). The darkest part indicates the genomic region that was inspected for heterozygous deletions. This rectangle is extended with a lighter coloured rectangle to indicate the actual span of the deletion (based on MLPA). Each of these 4 respective deafness patients have an ultra-rare hemizygous missense variant (M), indicated with a purple bullet.

conversions that affect the *OTOA* gene also lead to null alleles as a result of a LoF variant being introduced. This conversion that affects exon 22 of the *OTOA* gene (ENST00000646100) has previously been discussed by Laurent et al., 2014[8]. The conversion from *SMN2* to *SMN1* which was found in 6 patients with spinal muscular atrophy (SMA) is causative for disease as a result of a synonymous variant that is introduced in the *SMN1* gene. This variant leads to altered splicing and, as a consequence, results in a non-functional protein product[22]. By using MLPA we confirmed the bi-allelic losses of the *STRC* and *SMN1* alleles. Using long-range PCR and long-read PacBio sequencing we confirmed the bi-allelic losses of the *OTOA* alleles.

Importantly, among the individuals for which the deafness disease gene panel was not requested we did not identify any homozygous LoF-introducing gene conversion in *STRC* or *OTOA*. The same holds true for *SMN1*: all of the identified pathogenic gene conversions were exclusively found amongst SMA patients. This illustrates the very high precision of our calls (100%; p-value$_{\chi2,STRC}$ = 6.26e-2; p-value$_{\chi2,OTOA}$ = 1.51e-11; p-value$_{\chi2,SMN1}$ = 1.13e-11).

In total, the analysis of SNVs/Indels within paralogous coding regions of known disease genes in previously undiagnosed patients resulted in 14 new diagnoses.

### Systematic analysis of homozygous deletions results in 11 diagnoses

Analogous to the SNVs/Indels, homozygous deletions were filtered prior to clinical interpretation. Only events with a CAF ≤ 0.5% that

affect a gene that is present in the disease gene panel of interest for an undiagnosed patient were considered. Application of this filter resulted in 147 homozygous deletions (Supplementary Data 18). Among these were several known genetic causes for disease, such as a bi-allelic loss of *OTOA* exon 22 identified in two patients with deafness[8], and 9 homozygous deletions of *SMN1* exon 7 in SMA patients[23] all of which were confirmed with MLPA.

In the group of individuals for which the deafness disease gene panel was not requested, no homozygous *OTOA* deletions were identified. Among the individuals for which *SMN1* was not present in the disease gene panel of interest, only one homozygous *SMN1* deletion was found. This may represent a case with a very mild phenotype as has been reported in literature[24]. When we conservatively assume that this call is false positive, the precision of our *OTOA* and *SMN1* deletions remains high (91.7%; p-value$_{\chi2,OTOA}$ = 9.02e-7; p-value$_{\chi2,SMN1}$ = 3.51e-16).

Overall, the analysis of homozygous copy number variants in known disease genes revealed 11 pathogenic deletions leading to a diagnosis in previously undiagnosed patients (Table 2; Supplementary Figs. 7 and 8).

### Distinguishing ectopic gene conversions from deletions

By using *STRC* as an example, we wanted to investigate whether any patient was diagnosed with a pathogenic deletion but in which the real underlying genetic event is most likely an ectopic gene conversion (Fig. 1e, f). Our in-house diagnostic pipeline identified 58 homozygous deletions in the subcohort of patients with hearing impairment. All of these events were confirmed by using MLPA. Using Chameleolyser we also found homozygous losses of *STRC* alleles for these 58 patients. However, in only 37 of these, we actually detected a homozygous deletion. In the remaining 22 (37%) we identified, based on coverage profiles, a homozygous gene conversion from *STRCP1* to *STRC* (Supplementary Data 19). All of these gene conversions are predicted to affect at least exons 19-23 (ENST00000450892) and therefore introduce LoF variation into *STRC*. As a consequence, the pathogenicity of the identified deletions and gene conversions is the same and thus, the genetic diagnosis of a homozygous *STRC* deletion in the 22 patients in which we identified a gene conversion does not pose an issue. Nevertheless, the ability of Chameleolyser to distinguish homozygous deletion events from gene conversion events is clinically very relevant since the vast majority of gene conversions is benign. For example, in our cohort of 41,755 samples we identified 47 homozygous gene conversions from *STRCP1* to *STRC* that do not introduce LoF variation (Supplementary Data 20). Since these events are present in patients with all kinds of different phenotypes as well as in healthy parents of patients we can reasonably assume that these events are benign. This can only be true in case the alleles are indeed converted and not deleted. However, we note that using ExomeDepth all of these events are called as homozygous deletions of *STRC* exons. This potentially poses a risk for making an erroneous molecular diagnosis.

## Discussion

We developed a bioinformatics method to systematically analyse all coding paralogous regions in 41,755 individuals using existing WES data. We identified an average of 60 variants per sample that could not be detected using standard WES analysis. Of these, about 1% is a missense or LoF variant with an allele frequency ≤0.5% in one of the 332 OMIM disease genes that are affected by sequence homology (Supplementary Data 21). We carefully interpreted a subset of these variants, namely the variants within the genes in the requested disease gene panels. By doing so, we could establish a genetic diagnosis for 25 previously undiagnosed patients by either SNVs/Indels, gene conversions or CNVs, or the combination thereof. All of these pathogenic variants were identified in 1 of 3 genes: *STRC*, *OTOA* and *SMN1*. For the respective patient groups (patients with hearing impairment and patients with spinal muscular atrophy) our method solved >1% of

previously undiagnosed patients. As our approach identifies causal variants in known disease genes, we believe that it may also be used to find novel disease genes.

We noted that using standard data analysis approaches, CNV callers that are applied on WES or WGS data are unable to discriminate between gene conversions and deletions. Indeed, gene conversions are falsely called deletions as a consequence of a reduced number of reads at the acceptor site of the conversion (Fig. 1). Sometimes, the pathogenicity of a gene conversion and the corresponding deletion is the same, e.g. a LoF-introducing gene conversion in *OTOA* or *STRC*. In such a scenario there is no risk in making a wrong molecular diagnosis, and only the exact genomic alteration that is responsible for the patient's phenotype will be wrong. However, most gene conversions are benign, and a genomic deletion may be inferred by standard WES tools, where Chameleolyser could provide an accurate diagnosis. Our technical validation efforts demonstrate that a large part of the issues related to the analysis of duplicated genomic sequences are resolvable with novel sequencing technologies (roughly 90% of called variants are concordant with HiFi PacBio data). Undoubtedly, the generalised usage of these novel technologies will further help the field to characterise these difficult genomic regions—much beyond what Chameleolyser can offer based on short-read data. It also provides input to generate a more complete and higher quality human reference genome (T2T[25]) which in turn improves variant discovery for both short and long-read data[25,26]. We foresee that these feedback loops will continue to accelerate the quality of sequence analysis in the next decades. Currently, however, these novel sequencing technologies are highly expensive and therefore not affordable for a health-care system. As a result, the rate at which short-read data is produced is still much higher as compared to LRS. For all of these short-read data, our method offers an effective way to query the difficult parts of the exome and genome. In this study, we applied Chameleolyser to the large number of WES datasets that are currently available in our medical genetics center[27,28]. Chameleolyser could equally well be applied to short-read WGS data. Direct application would however only consider homologous coding regions. A future update of Chameleolyser for WGS could also incorporate homologous regions that only affect non-coding regions, although the interpretation of identified variants in such regions would be very challenging.

In conclusion, we present a bioinformatics method to identify genetic variation in paralogous genomic regions. By analysing 41,755 WES samples we identified a genetic diagnosis in 25 previously undiagnosed patients. We expect that Chameleolyser can substantially contribute to future discoveries based on genome variation that has so far remained hidden.

## Methods
### Samples
The analysis was applied on 41,755 WES samples including 6980 patient-parent trios (20,940 samples (50%)). All samples were sequenced either using Illumina HiSeq2000, Illumina HiSeq4000 or BGI DNBSEQ short reads sequencing platforms. Six thousand eight hundred and ninety-four exomes were enriched by using the Agilent SureSelect Human All Exon V4 kit while for the other 34,861 Agilent SureSelect Human All Exon V5 was used. All of these data were generated between 2002 and 2020 as part of the routine genetic investigation from Genome Diagnostics Nijmegen. As such, data processing was conducted with our standard diagnostics WES pipeline[21]. Depending on the patient's phenotype a specific gene panel is requested in which genetic variants are inspected and interpreted. After this diagnostic screening, 17,650 patients remained molecularly undiagnosed (Supplementary Data 14).

### Identification of paralogous regions
Two different sets of paralogous regions were derived after which they were merged. For the first set, all protein-coding genes with one or multiple pseudogenes were used as a starting point. For the second set, the genomic coordinates of reads with low mapping quality were used.

**Set I: Regions in protein-coding genes with known pseudogenes.** Starting from all pseudogenes in the comprehensive gene annotation file from Gencode 31 (lift37; https://ftp.ebi.ac.uk/pub/databases/gencode/Gencode_human/release_31/GRCh37_mapping/gencode.v31lift37.annotation.gff3.gz), those with a corresponding protein-coding gene were selected (n = 1680). This correspondence was based on the HGNC gene name in the file. Next, by using MAFFT v7.407[29], a multiple sequence alignment was generated between each protein-coding gene and its pseudogenes. To only keep the regions for which the protein-coding gene has exactly one paralogous pseudogene and for which the sequence identity between the protein-coding gene and the pseudogene is 90% or more, a sliding window approach was used (window length = 100 bp). This resulted in a set of regions in 989 protein-coding genes with their respective pseudogenes (Supplementary Fig. 9).

**Set II: Regions corresponding to low mapping qualities.** For 250 randomly chosen WES samples (SureSelect Human All Exon V5), low-quality reads (i.e. mapping quality (MQ) < 10) were extracted using samtools 1.9[30]. To avoid that poorly covered regions (and thus uninformative in terms of variant identification) are included in the region set, bedtools v2.28.0[31] was used to only keep the genomic positions with a sequence depth ≥ 10. Because many regions were fragmented, i.e. separated by only a few bases, the resulting regions were merged using three different slopping distances: 250 bp, 500 bp and 5% of the region length (bedtools). In the last list, we removed regions <50 bp (Supplementary Fig. 10a). To find the homologous relations between the regions in the lists of regions we performed pairwise sequence alignments (PWA) with EMBOSS Needle v 6.6.0.0[32] as follows. Per list (250 bp, 500 bp and 5%), PWAs were made between each region and each other region in the file, as well as with its reverse complement (Supplementary Fig. 10b). We then defined sets of paralogs in the following way: all regions within a set should mutually have an alignment score ≥0.9 (it has been shown that the sequence identity between a linked donor and acceptor site is ≥90% with very few exceptions[33]). Furthermore, paralogs are only tolerated in a set when they do not have an alignment score ≥0.9 with a region in another set of paralogs (Supplementary Fig. 10c). Before merging the 3 lists (each corresponding to a different slopping distance), sets without any exonic overlap and with >5 members were removed. The actual merging process starts with the region list corresponding to a slop distance of 500. To that list, regions from the 2 other lists were added (first 250 bp, next 5%) with the following rule: only if a region does not overlap with regions which are already in the merged region list, the region is added. This resulted in 1334 regions with their paralogs.

**Merging region set I and II**. To combine region set I and II into one final set of regions to operate the paralogy analysis on, the same iterative procedure as above (i.e. merging set II regions with different slopping distances) was used. Here, we start from the full list of set I regions. To that list, set II regions were added only if there is no overlap. Doing so, 177 sets of regions were removed. The output of this step corresponds to the variant calling regions in the paralogy analysis (Supplementary Data 23). In order to limit the number of broken read pairs in the extraction procedure (cf. mapping and short variant calling), the region list for read extraction consists of the same regions extended with 500 bp up and downstream (Supplementary Data 24).

**Generating the masked reference genome.** When re-aligning the sequencing reads, all regions except one in a set of paralogs should be masked in the reference genome. For this, we choose to mask the

regions that had the least overlap with a protein coding sequence (Supplementary Data 25). Masking was conducted with bedtools v2.28.0.

## Mapping and short variant calling

All reads overlapping the read extraction regions were extracted using samtools 1.9. On the resulting alignments, variant calling using GATK 4.1.2[34] was performed. Next, the aligned reads were converted to FASTQ using samtools 1.9. BBmap v 38.56 was used to remove broken read pairs. Alignment of the reads to the masked reference sequence was done with bwa 0.7.17[35]. To remove duplicate reads, Picard v. 2.20.8 was used. Because reads from paralogous regions are aligned to a single region, we do not expect 50% allele ratios for variants. Therefore we used LoFreq 2.1.3.1[36] for sensitive variant calling on this newly generated alignment (parameters: no-default-filter, use-orphan, no-baq, no-mq, sig = 1).

## Identification of deletions and conversions

**Determination of subregions.** Region set I contains regions that consist of multiple exons and introns. Since we want to call deletions and conversions at the resolution of a single exon, we split these regions into subregions. This was achieved by intersecting the list of regions with the list of all protein-coding exons from Gencode 31 (including 200 bp intronic flank; lift37). For region set II this operation is not needed because these regions are small by design (i.e. they do not contain large introns). Nevertheless, in order to ensure high-quality deletion and conversion calls, region set II was filtered to only keep sets consisting of 2 paralogs. In total, we end up with 4921 subsets of regions (Supplementary Data 26).

**Read counts per subregion.** Bedtools v2.28.0 was used to derive, per sample, the number of reads that align to the different subregions. This coverage calculation was applied on the original alignment (no masking applied).

**Kernel density estimation.** In order to accurately identify deletions and ectopic gene conversions it is important to consider a large set of samples at once. Doing so, it can be seen from the coverage distribution whether a poorly covered sample is part of a wide distribution originating from samples without deletion or it is not part of such a distribution and thus it is likely a sample with aberrant coverage due to a genetic event. The identification of these coverage peaks is done in 2 steps. First, we estimated the density of the data with the technique of kernel density estimation (KDE). This technique smoothens the discrete data, i.e. it results in a continuous curve which aligns with the density of the data. For this scikit-learn was used (exponential kernel; default parameters)[37]. After having estimated the density we applied the argrelextrema function from the scikit-learn software package to determine the local minima and local maxima of the curve. This results in peaks or KDE clusters. This operation was separately done for deletions where both paralogs in a set were deleted and deletions plus gene conversions affecting only 1 region in a set.

**Deletions of both paralogs within a set of paralogs.** To detect the events where all regions in a set of paralogs are deleted, a vector with the per-sample number of reads that align with one of the regions in the set is used as the input for the KDE. This is only done for sets for which the median (in the cohort; per enrichment kit) of the total number of reads is 120 or more. To find rare deletions, all KDE clusters (peaks) having >10 reads or corresponding to >10% of samples in the cohort were excluded (taking into account that some read ends might have an alignment in the deleted region, we tolerate some reads to be aligned). The analysis was done separately for Agilent V4 and Agilent V5 samples and, importantly, for this analysis, both uniquely and non-uniquely aligned reads were used. This resulted in 1962 calls in our full cohort.

**Gene conversions or deletions of one region in a set.** To identify gene conversions or deletions of one region in a set of regions, a vector of per-sample read count ratios is used. We call this ratio $R$: for a set of paralogs consisting of region X and region Y it is the number of reads that uniquely align to X divided by the number of reads that uniquely align to X or Y. To be able to technically discriminate between a low (possible acceptor) and a high (possible donor) covered region, within this vector, samples with <30 uniquely aligned reads (sum of both paralogs) were excluded. Furthermore, regions for which the median number of reads (in the cohort; per enrichment kit) in one of the two paralogs is below 60 were excluded from the analysis (To be able to discriminate samples with low from samples with high coverage). To identify rare genetic events, all KDE clusters (peaks) corresponding to >10% of the samples in the cohort were excluded for further analysis. Furthermore, all alterations not overlapping a protein-coding gene were discarded. This resulted in 49,151 calls. Within this call set we defined an event to be either the homozygous deletion of X or the conversion from Y to X if 1) a maximum of 10 uniquely aligned reads onto X 2) $R \leq 0.05$ (e.g. we tolerate (acceptor) regions with 10 reads aligned to them only if the number of reads on the possible donor site is at least 20 times higher).

The distinction between deletion and conversion is based on the number of reads that uniquely align onto Y (the donor site in case of a conversion). First, the number of uniquely aligned reads onto X and Y are normalised per sample (based on the total number of reads over all paralogs). Next, for each sample, a one-sided percentile was calculated. For sample S and region X we call this metric PercDPN$_{X,S}$, for region Y it is PercDPN$_{Y,S}$. We now calculate a threshold

$$T_{X,S} = \text{PercDPN}_{X,S}^{\frac{1}{3(1-3\text{PercDPN}_{X,S})}} \tag{1}$$

An event is predicted to be a gene conversion if PercDPN$_{Y,S}$ < T$_{X,S}$ (2) and if the normalised number of reads that align onto Y > $\frac{4}{3}$ Median$Y_{norm}$ (3). If these conditions are not met, we predict region X to be deleted. Formula (2) is implemented to accommodate for the fact that not all ectopic gene conversions are very rare. The basic idea is to require a more extreme coverage (i.e. an extreme observation in the subcohort of all samples enriched with the same enrichment kit) on the possible donor site when the coverage on the acceptor site is very extreme. When for example 1000 samples only have a few reads aligned onto the possible acceptor site then it is possible and likely that there is at least in part of the samples an ectopic gene conversion. As a result, quite some samples will have an elevated coverage on the donor site. If we would then require a very extreme coverage observation in order to predict the event as an ectopic gene conversion, we would be wrong for most of the samples (overly conservative). When only a few samples have a very low coverage on the possible acceptor site, it is impossible that there are many samples with an ectopic gene conversion. So in that case we can be more strict. By replacing PercDPN$_{X,S}$ by a very small value (e.g. $10^{-5}$) in the formula, we can see that the formula can be approximated by the cubic square of PercDPN$_{X,S}$. When we on the other hand substitute the parameter by $10^{-1}$ it can be approximated by the square root and thus a less stringent read depth requirement for the possible donor site.

**Combining single exon CNVs.** After having derived the deletions and conversions per subregion we merged the ones that are in direct proximity. For this, all filtered calls were annotated with their overlapping gene name using Gencode 31 (lift37). First, the calls were combined per gene (e.g. if for a certain patient 2 different exons are deleted in the same gene, these calls are merged into 1 deletion). Next, deletions and conversions in neighbouring genes were merged. Also when 1, 2 or 3 coding genes (but not necessarily part of a paralogous set of sequences) exist between the 2 different CNV calls, it was

assumed that these originate from the same genetic event and therefore these calls were merged into a single call.

## Short variant processing

After having called variants in the original and newly generated alignments (i.e. after masking) we made a raw variant call set with variants of interest.

**Raw call set.** The raw variant call set consists of variants satisfying the following criteria:

– The variant is not an alteration of a singly unique nucleotide (SUN; i.e. sequence difference between homologs).
– Depth of overage at the position of interest in the masked alignment ≥60. With 30x read depth an almost optimal sensitivity is achieved in WES for SNV/Indel identification[38]. A threshold of 60x is chosen since the vast majority of paralogous sets consist of 2 paralogs.
– Variant allele fraction (VAF) is ≥0.15. In a pool of reads originating from 2 alleles, a heterozygous variant has an expected read ratio of 0.5. This ratio becomes 0.25 for a heterozygous variant in 1 of 2 paralogs, each having 2 alleles. If we consider all variants with a read ratio of 0.15 and higher, we obtain 97.9% sensitivity under a binomial model assuming 60x read depth (and probability 0.25). As shown before, the distribution of the ratio of reads that support the variant allele approximately follows a binomial distribution[39].
– The variant is not present in the original VCF (GATK variant calling; no masking).

This resulted in 56,156,453 calls.

**Expansion: from variant calls to variants.** Except for calls corresponding to homozygous variants in all paralogs, it is unknown in which region (within the group of paralogous regions) the variant is present. For that reason, we need to compute all possible variants (i.e. VAPs) corresponding to a variant call. The number of possible variants equals the number of paralogs in the set. The actual computation is done with an in-house Perl script and is based on the MSA between the different paralogs. The 56,156,453 calls correspond to 119,551,166 VAPs.

**Annotation.** All variants were annotated on canonical transcripts using Ensembl VEP 97.

**Filtering.** Several filters were applied in order to transform the raw variant call set to a high accuracy call set with variants of interest:

• Since the major focus is on (relatively) rare variants, we excluded variants which were observed in > 10% of the analysed samples for further analysis. This resulted in 4,064,684 calls corresponding to 8,753,075 VAPs.
• Variants with an apparent good quality in a particular sample but which are of low quality in most other samples were filtered out. This is implemented as follows: the variant is removed from the call set if the number of samples having the variant with a VAF > 0.05 is more than twice the number of samples having the variant with a VAF > 0.15. With that we filter out variants for which most samples have a VAF between 0.05 and 0.15. This resulted in 2,474,765 calls corresponding to 5,152,554 VAPs.
• It has been shown that the Illumina sequencing technology is prone to small indel errors within homopolymer tracts[40] and post-homopolymer substitutions[41]. In general, the longer the mononucleotide run, the more sequencing bias is introduced, but from a tract length of 6 and more, the errors become most apparent. For that reason, we took genomic coordinates for all homopolymers in the genome >5 mononucleotides from https://github.com/ga4gh/benchmarking-tools. These intervals were

extended up and downstream with 2 base pairs. All variants in these regions were excluded for further analysis. This resulted in 2,337,271 calls corresponding to 4,859,954 VAPs.
• We excluded variants in subregions containing 5 singly unique nucleotides in a stretch of 10 bp or less. We did so because the pairwise alignments of paralogous regions (which was used to derive the singly unique nucleotides) and the alignment of the short next-generation sequencing reads (based on which variants were identified) can be slightly different in regions with several sequence differences between paralogs. If we would not ignore these subregions we would have inflation of false positive variants due to unrecognized singly unique nucleotide. This filter step reduced the variant call set to 2,191,707 calls corresponding to 4,596,461 VAPs.

## Validation

**Technical validation.** HiFi sequencing reads were generated for 20 samples using the Pacific Biosciences Sequel II instrument with Chemistry 2.0. DNA for all samples was sheared using a Megaruptor 3 instrument aiming fragments of 18 kb. SMRTbell Express 2.0 was used to prepare the library, the PippinHT instrument for fragment size selection >10 kb. Finally, sequencing was conducted with 3 SMRT Cells per sample targeting 30x coverage. Sequencing reads were aligned to the GRCh38/Hg38 genome with minimap2[42]. Variant calling was conducted with DeepVariant[17].

Chameleolyser initially identified 15 homozygous deletions, 8 homozygous gene conversions and 847 SNV/Indel calls not due to a gene conversion within these 20 samples based on the short-read data. The coordinates of these variants were converted to hg19 with CrossMap[43]. For SNVs/Indels not due to gene conversion, if only one possible variant could be converted (and thus the other(s) failed to be converted) we discarded the variant for validation purposes. By doing so, we ended up with 769 variant calls to be validated. Deletions and gene conversions were manually checked using the Integrative Genomics Viewer (IGV). For SNVs/Indels (not due to a gene conversion), a 2-steps approach was followed. Firstly, the variant was checked in the VCF. This resulted in 557 calls that were concordant with the long-read data. Next, we manually checked (using IGV) the variants that could not be validated using the VCFs. This resulted in an extra 121 validated variant calls (Supplementary Data 5).

All 15 deletion calls were visually inspected in the LRS alignments. For 8/15 deletion calls there is a maximum of one LRS read aligning onto the region that Chameleolyser claims to be deleted (as opposed to samples without the deletion). For 5 of the remaining 7 deletions the reads that align onto the region corresponding to the deletion call are all read tails with a large number of non-matching bases. Read tails were blasted to the reference genome, showing that these read portions actually correspond to the region up or downstream of the actual deletion (Supplementary Fig. 11). The two remaining deletion calls are in complex regions (the *KIR* gene cluster) and we could not unequivocally come to the same conclusion. By comparing the alignments with samples without the deletion call we found that there is a genetic event, but not necessarily a homozygous deletion. For that reason, we conclude to have 13/15 deletion calls which correspond to LRS alignments.

All conversion calls were visually inspected in the LRS alignments for the absence of a deletion or coverage drop. In addition, we checked for homozygous variants that correspond to the sequence differences between the linked donor and acceptor site. We confirmed this for all 8 ectopic gene conversion calls.

We downloaded exome and PacBio LRS data from https://github.com/genome-in-a-bottle/giab_data_indexes for 5 genome-in-a-bottle (GIAB) samples (HG002, HG003, HG004, HG005 and NA12878). The SNVs/Indels (not the result of gene conversions) were validated with the same procedure as described above for in-house validation samples.

**Trio-validation.** We considered all genomic sites with a minimal read depth of 60 in the masked alignment. For these sites, we counted the number of variants in the child for which the variant allele fraction in the mother and in the father is below 1%. These variants were considered de novo. If, on the other hand, the variant was identified in the father or mother (thus having a variant allele fraction above 15%), we considered this variant as inherited.

**Comparison with other variant callers**
**SNVs/Indels (not the result of gene conversions).** GATK is run within the Chameleolyser method (cf. Mapping and short variant calling). DeepVariant 1.5.0 was run in a Docker container as described in the readme (https://github.com/google/deepvariant). To calculate the sensitivity of Chameleolyser, GATK and DeepVariant for SNVs/Indels (not the result of gene conversions) we first derived a set of high-quality SNVs/Indels. To do so, we applied DeepVariant on 25 samples for which 30x High-Fidelity LRS alignments were available (20 in-house + 5 GIAB). All variant calls with a quality score > 30 which are present in the homologous regions that are used in this study (cf. Identification of paralogous regions) were considered true positive. Since all LRS samples were aligned to hg38 and exome data were mapped against hg19, a liftover of the coordinates was needed. This was done with CrossMap[43]. Because some of these true positive genetic variants might be present on genetic sites that are not or insufficiently covered in the exome experiment, we only considered variants for sensitivity analysis if the read depth for the corresponding base in exome data is ≥ 20. A variant is considered to be present in a zero mapping quality region if all reads that cover the respective position have mapping quality = 0.

**Homozygous deletions and gene conversions.** ExomeDepth and Conifer were applied to all exome samples in the study cohort. For ExomeDepth, capture target files were subdivided according to Parrish et al. 2017[44]. Reference pools were created each consisting of 500 samples from healthy sex-matched individuals which were sequenced on the same sequencing machine and for which exome capture was done using the same enrichment kit. For Conifer, the same initial approach for reference pool selection was used. Here, bad-quality reads (average quality score < 20) were removed from the samples. Next, the standard analysis steps were undertaken for both ExomeDepth and Conifer. All deletions in the output of ExomeDepth that overlap with a paralogous region and that have an observed/expected read ratio < 0.1 were considered for further analysis. For Conifer, deletions with a SVD-ZRPKM ≤ −3 were selected. Next, in analogy with Chameleolyser (and thus for comparability), we removed all deletion calls with a cohort frequency > 10%. The remaining deletions that were present in the 20 samples for which we generated LRS data were used for the comparison between the CNV callers (Supplementary Fig. 4). Sensitivity could not be assessed for deletions and gene conversions for two reasons. 1) we cannot derive a set of true positive (or approximated by high-quality) events with a cohort frequency ≤ 10% because we only have a limited number of LRS samples available. This would be needed because the CNV calling within Chameleolyser is restricted to events ≤ 10% which is part of the method itself. 2) There are no tools to identify ectopic gene conversion events from LRS data. To estimate the number of true positive events among the deletion calls from ExomeDepth we first converted the regions from hg19 to hg38 by CrossMap. Only 151 regions could unambiguously be converted. If the number of LRS reads in the complete region corresponding to the deletion call is 20 or less, we presumed the region to be deleted. The reason for allowing 20 reads is that for the deletions which were manually validated by visual inspection up to 20 reads can be present in the deleted region as a result of suboptimal alignment of read endings (Technical validation, Supplementary

Fig. 11). Nevertheless by choosing a different threshold, the difference between Chameleolyser and ExomeDepth remains the same.

**Variants of clinical interest.** OMIM disease gene annotations were fetched from Ensembl Version 97 (MIM morbid 12/04/2019).

**Variants with ambiguous positions.** Exome-wide VCF files for all 41,755 WES samples were available through our in-house diagnostic pipeline (Radboud University Medical Center). This includes read alignment with bwa-mem 0.5.9-r16 and variant calling with GATK 3.2–2. In order to only retain high-quality variants, we filtered variants based on GATK's quality score: only substitutions with a GATK quality score ≥ 300 and indels with a quality score ≥ 1000 were taken into consideration[45,46].

**P value calculations.** The p-values in the paragraphs 'Validation' and 'Variants with ambiguous positions' are derived from a binomial test in R (two-sided). All other statistical tests in this manuscript are chi-squared tests, corrected for multiple hypothesis testing per series (Bonferroni's method; R 3.5.1).

**Reporting summary**
Further information on research design is available in the Nature Portfolio Reporting Summary linked to this article.

## Data availability
The validation data generated in this study have been deposited in EGA under accession codes EGAS00001006479 (long-read genome sequencing for individuals with biobank consent (https://ega-archive.org/studies/EGAS00001006479)) and EGAS00001007513 (STRC amplicon sequencing (https://ega-archive.org/studies/EGAS00001007513)). These datasets are available under restricted access. Re-use of the data will be evaluated by a data access committee whether the proposed re-use is in line with the consent. Supplementary Table 1 describes the mapping between the EGA sample identifiers and the identifiers that were used in this manuscript. The data onto which Chameleolyser is applied in this study is collected through routine genetic investigation. A diagnostic laboratory can use (de-identified) samples from archived clinical samples to validate and implement novel diagnostic assays. The derived clinically relevant variants can be shared, but in the absence of explicit data-sharing consent at the individual patient level, complete FASTQ, BAM and VCFs cannot be disclosed unless specifically consented to by individual patients. These methods are also in accordance with relevant guidelines and regulations and approved by the institutional review board of the Radboud University Medical Center (2020-7142) and the Declaration of Helsinki. Source data are provided with this paper. The processed data that support the findings of this study are available as Supplementary Data 1–26. The genome-in-a-bottle data used in this study are publicly on NCBI (URLs available on GitHub (https://github.com/genome-in-a-bottle/giab_data_indexes)) and/or the PacBio cloud (https://downloads.pacbcloud.com/public/). A list of download URLs per sample is also available as Supplementary Note 1. Source data are provided with this paper.

## Code availability
The tool (Chameleolyser) as well as all other code that was used to produce tables and figures is available on GitHub (https://github.com/Genome-Bioinformatics-RadboudUMC/Chameleolyser)[15].

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

## Acknowledgements

The aims of this study contribute to the Solve-RD project (to H.B., A.H. and C.G.) which has received funding from the European Union's Horizon 2020 research and innovation programme under grant agreement No 779257.

## Author contributions

W.S. conceptualised and designed the study, conducted the data analysis and wrote the original draft. L.H-W., R.P., E.d.B. and A.S. did the clinical interpretation of variants. J.H. contributed to the figures. D.H. and M.S. did the wet-lab validation of variants. E.K., H.Y., H.B., A.H. and C.G. reviewed and edited the original draft. H.B., A.H. and C.G. supervised the project.

## Competing interests

The authors declare no competing interests.
