## [Peer Review File · Nature Communications]

Systematic analysis of paralogous regions in 41,755 exomes uncovers clinically relevant variationReviewers' Comments:

Reviewer #1:

Remarks to the Author:

In this manuscript the authors have developed a novel method (Chameleolyser) to identify genomic variants in paralogous regions of the genome which are intractable to standard analysis methods. They have applied it to a large cohort of 41,755 whole exome sequencing (WES) samples, 50% of which are in the form of trios. The cohort were sequenced as part of the routine genetic investigations at Genome Diagnostics Nijmegen.

They demonstrate that this method is able to identify a huge number of rare homozygous deletions and SNVs and they are able to show that a number of these are the result of gene conversion events. The quality of these variants was confirmed using complimentary long-read sequencing of 20 samples.

In total, the authors state their method has resulted in 25 molecular diagnoses in patients that were previously undiagnosed.

This manuscript represents a substantial piece of work and a novel approach to interpreting the genome of patients to find novel diagnoses in patients presently undiagnosed. The concept of exploring paralogous regions of the genome is interesting but the methodology needed to do it is initially difficult to fully understand. I believe this type of analysis will be of interest to a wide range of people, from those interested in functional genomics through to those in clinical genomics.

There are however a number of points I would recommend need to be addressed before the manuscript is ready for publication.

Firstly, and in a broad sense, the manuscript needs to better explain the concept of how the method works, in particular how paralogous regions (both 100% identical and <100% identical) are masked and the differences between homozygous deletions and gene conversions. Figure 1 doesn't provide a clear enough description of this and as this is fundamental to understanding the methodology I would like to see this made clearer to the reader.

I also find it strange that there is almost nothing written about the software package Chameleolyser, what language is it written in, what is the input file format, how long does it take to run, what is the output file format etc. There is a link to the GitHub page but I believe it will be of great importance for people to know whether they have the computing infrastructure to run this tool, the correct file formats and an idea of how long the analysis takes. If it takes a day to run a single sample then it will probably not be applicable to other large cohorts without a serious commitment of resources.

And finally, although every new diagnosis is hugely important to that patient, only 25 new diagnoses were made from 17,650 undiagnosed patients (0.14%). This shows that this method must be super easy and quick to run and interpret if it is to be incorporated into routine genomic testing.

Below is a list of questions/comments that highlight specific areas that I feel need to be explained in more detail to make the manuscript more accessible and understandable:

1. Line 49 – You mentions CNV callers in the paragraph but highlight figure 1e which represents a SNV, this is confusing.
2. Figure 1c, why is R2 not masked?
3. Line 57 – you mention methods to identify CNVs and say a limited number are specifically designed to estimate copy number of paralogous regions but do not reference them, please add in refs.
4. Can you explain why you only look at homozygous deletions or gene conversions, it's not immediately clear to me why heterozygous events cannot be detected.
5. Results section, you give the total number of variants identified but could you give a per sample

overview.

6. Validation – what was the criteria for choosing the 20 samples that underwent long-read sequencing, was it random or based on something else?
7. Identification of paralogous regions – can you reference where you obtained the lists of paralogous regions.
8. Line 278 – what does the software MAFFT do?
9. Line 291 – what does EMBOSS Needle do?
10. Kernel density estimation and peak filtering – can you explain in more detail what KDE is and why scikit-learn was used. Also reference them.
11. In line 379 you use a frequency value of VAF and in line 398 you use percentage, please chose one and use throughout.
12. Line 423 – change ‘calls to were’ to ‘calls that were’.
13. Line 425 – what is meant by the term ‘automated fashion’?

Reviewer #2:

Remarks to the Author:

The authors present a tool Chameleolyser, designed for variant calling in duplicated regions based on Whole-Exome Sequencing (WES). The authors address a medically-relevant problem, present a novel solution, validate results using long-read sequencing, and use it to diagnose 25 previously undiagnosed patients. The paper has several deficiencies, but the method fulfills its main purpose: to identify potential disease-causing mutations in the duplicated genes.

Major: the authors do not benchmark Chameleolyser against other CNV and variant calling methods. In duplicated exons, Chameleolyser is able to detect (i) large deletions, (ii) sequence variants, (iii) gene conversion events. Even though copy number variation detection using WES is not as advanced as with WGS, there is a number of CNV-detection tools that use WES data (Zhao et al., BMC Bioinf., 2020). The authors should compare against these methods to show that Chameleolyser performs better than existing tools.

The authors should also compare their variant calling performance against other short variant callers, for example GATK, Freebayes and DeepVariant. Even if these tools are not designed for duplicated regions, the authors need to prove that their method works better than existing variant callers.

Major: Homozygous SNVs can be identified using standard variant calling (as they would appear on all repeat copies irrespective of read mapping). The authors should show that their method better identifies homozygous variants than standard variant callers.

Major: 8/8 gene conversion events and 11/15 deletion events were validated (line 99). The authors write that the four remaining deletion events could not be validated as LRS alignments were unclear. However, the validation was performed manually (line 422). Consequently, the authors should describe, what are their criteria for low LRS clarity, and how many of the remaining 8 gene conversions and 11 deletions have low LRS clarity.

It is not required, but the paper would benefit from higher sample size. In order to achieve that, the authors may analyze samples with publicly available WGS/WES data and PacBio HiFi data, such as a subset of samples from the 1000 genomes project. Alternatively, the authors can analyze 7 samples, collected by the Genome in a Bottle consortium (GIAB), that have high-confidence variant call sets, which can be used in an additional validation step.

Major: Authors rely on singly unique nucleotides (SUNs, also known as paralogous-sequence variants, or PSVs) to identify gene conversion events. However, it has been shown before (Sudmant et al., Science, 2010; Mueller et al., Am.J.Hum.Genet., 2013), that many SUNs are polymorphic in the population. In this case, relying on such SUNs can lead to incorrectly identified gene conversion events.

Major: The methods are ad-hoc, with many arbitrarily selected thresholds and parameters (for example, lines 287, 295, 341-2, 349, 351-5, 361-3, 378-9, 394, 398-9, 401, 405, 438). The authors should aim to reduce the number of parameters and explain how parameter values were selected where possible.

Minor: The method identifies disease-causing mutations in just a handful of well-known genes (STRC, OTOA, SMN1). Is the method applicable in other disease-relevant duplicated genes?

Minor: Are there any estimates of the method sensitivity, i.e. the percentage of variants that are present in the data and found/not found by Chameleolyser? If the method has low sensitivity, explain why it is not important in this case.

Minor: The authors seemingly use the term "SNV" for both short mismatches and indels, and use term "deletion" for gene/exon deletion events. In my opinion, this is a bit unclear. Perhaps, the authors can use terms "sequence variants", "short variants", "SNVs and indels", etc. to better underline the difference between two concepts.

Minor: Line 85: 2,191,707 SNVs. Does this number include the same heterozygous variant split into several positions? If yes, the number is misleading. Same on line 146.
Line 388: True number of variants lies between 56M and 119M. Saying that there are 119M variants is misleading. Same goes to lines 394-407, perhaps some other terminology or abbreviation can be used.

Minor: Line 108-...: explicitly mention if SUNs are discarded, or considered as VAPs?

Minor: Line 125: better to use terminology "p-value" than simply "p".

Minor: Line 128: mention, that in non-duplicated (unique) regions. Also, the next sentence may be a bit easier to read, if it would be fractions/percentages instead of amounts relative to synonymous variants.

Minor: Line 140: "We filtered for" may be better to replace with "We selected" or other phrase, otherwise it is ambiguous (can be "we discarded").

Minor: Line 294: What happens if a paralog has similarity > 0.9 with some paralogs in the set, but < 0.9 with other paralogs? Line 296: what about regions that have similarity > 0.9 with two sets?

Minor: Line 319: Provide more description about GATK and LoFreq variant calling. Do the authors use GATK on the input WES data, and LoFreq on the reads, re-mapped to the masked genome?

Minor: Line 329: Why only 2-copy regions are analyzed? Does the method work in higher-copy duplications, for example NCF1, SRGAP2, etc.?

Minor: Line 336: Need more details about KDE, what parameters are used, if there is any post-processing.

Minor: Line 370: Specify merge distance or criteria.

Minor: In the results (line 105), the authors say that the 99% of SNVs present in the offspring are called in one of the parents. Does this mean that in methods (line 430) that the variant in a child and in a parent have VAF $\geq 15\%$? Does this include very common variants? If yes, the authors can show the numbers for rare variants as well.

Reviewer #3:

Remarks to the Author:

The paper by Steyaert et al. is a valuable contribution to the field, as it shows what I believe are clear examples where they were able to genetically explain and diagnose patients that had previously gone without a genetic explanation, by identifying variants in paralogous genes where short-read sequencing data cannot accurately align reads.

All of my suggestions are very easily addressed, so I classify them all as Minor, but many of them are very important to ensure readers accurately interpret these important results. I sincerely hope these suggestions are well received, as I only intend them to be constructive and to ultimately improve the work and paper.

Sincerely,

Mark T. W. Ebbert

Minor:

1. Clarify that unique sequence to identify ectopic events is generally only necessary for short-read sequencing technologies because long-read technologies will generally have reads long enough to anchor within adjacent unique sequence.
 - a. Suggested edit: "In order for ectopic gene conversion event to be detectable using short-read sequencing technologies, the donor and acceptor site should differ in at least one nucleotide."
2. Correct definition of ectopic gene conversion. Authors state that ectopic conversion events are copied from one genomic region to "a distant homologous region" (i.e., the acceptor site). The acceptor site itself *could* be homologous, but does not have to be, and often is not.
 - a. Suggested edit: Remove the word "homologous".
3. Transition between sentences doesn't quite follow, logically. In one sentence, the authors state that the two paralogous regions must differ by at least one nucleotide to be *detected* (for short-read technologies) and the following sentence uses this as justification that these events introduce new genetic variation. It's true that these events introduce genetic variation, but the logic is a bit disconnected.
4. Figure 1e. Maybe I'm missing something, but I do not understand why the authors state that reads from R1 (what I believe is the acceptor site) would only align to R2 (what I believe is the donor site). Specifically, I do not understand what the authors are saying in the third paragraph of the introduction. In paragraph two, the authors are talking about the paralogous regions as though they have unique sequence. If they have sufficiently unique sequence, which Figure 1e suggests, then why would reads from R1 only align to the donor site (R2)? If the R1 is sufficiently unique and the paralogous genes are included in the reference genome, then the aligner would certainly align R1 reads to R1. If R1 and R2 are *not* sufficiently unique, then the aligner would assign the reads a mapping quality of zero (0) and randomly assign the reads to both R1 and R2. Figure 1e clearly shows reads from R2 aligning to R2, but there are zero R1 reads present anywhere. i.e., R1 reads are not aligning to R2 in Figure 1e as paragraph 3 suggests they would (though I don't see why they would).
5. Because the nuances of this work are so critical, the authors need to explicitly state when they're talking about variants in the patient's DNA vs differences between genomic regions. e.g., I'm intimately familiar with these nuances, but even I had to re-read paragraph 4 of the introduction to understand what the authors were referring to.
6. At risk of being self-serving, while many aspects of the authors' algorithm are unique, they also use many of the same methods we described in our 2019 paper but did not cite our paper. For example, the authors specifically mention Ebbert et al. 2019 in the introduction, but do not explicitly cite the paper. Additionally, they do not give credit in the methods or results where they probably should.
7. Figures are low quality in the draft I received.
8. "patients dna" in Figure 1 should be "Patient's DNA"

9. Many of the terms the authors use are ambiguous. e.g., the title for table 2 is unclear whether the authors are claiming they discovered a new pathogenic variant that has never been associated with disease, or that the variants themselves were known to be pathogenic, but hadn't been identified in the patients.

10. Table 2 is also missing critical information on the variants discovered. Authors should include the chromosomal position. In the case of a deletion, they should specify exactly what is deleted.

11. IGV screenshots showing examples of some of these events would be very helpful for readers.

12. In some areas of the Results, it's difficult to know whether the authors are claiming new variant associations, or identifying known pathogenic variants in undiagnosed patients. The authors should be super clear about this in every instance.

13. The authors should distinguish between evolutionary gene conversion events and de novo events, where appropriate.

**Response to referees**

We thank the reviewers for their constructive feedback on our manuscript. We have addressed all of
the reviewer's concerns in a point-by-point response below.

**Reviewer #1 (Remarks to the Author)**

Firstly, and in a broad sense, the manuscript needs to better explain the concept of how the method
works, in particular how paralogous regions (both 100% identical and <100% identical) are masked and
the differences between homozygous deletions and gene conversions. Figure 1 doesn't provide a clear
enough description of this and as this is fundamental to understanding the methodology I would like to
see this made clearer to the reader.

We would like to thank the reviewer for his/her comment. We agree that the explanation of our method
could be improved. To address this, we now carefully explain the method in the results section. The
first 2 paragraphs of the results section now become (modified text in italics):

*Chameleolyser works by extracting reads in the 3.5% of the exome that is affected by sequence
homology (paralogous regions (Methods)) and re-aligning them to a reference genome in which all
but one paralogs within each set of paralogs are masked¹². By masking all nucleotides in these
regions in the reference genome, no sequencing reads will be aligned onto them. As a result, all
reads that originate from a set of paralogous sequences are uniquely aligned onto a single region in
the reference genome (the non-masked region; Figure 1b). Subsequently we perform sensitive variant
calling to identify SNVs (Methods).*

*Homozygous deletions and ectopic gene conversion events are identified by analysing the coverage
profile in the original alignment (without masking). In short-read sequencing data, a homozygous
deletion and the acceptor site of a homozygous ectopic gene conversion appear identical: no reads are
aligned onto that site of the reference genome. By also considering the number of reads that align onto
the paralogous regions, it is possible to discriminate between deletions and gene conversions. In case
of an ectopic gene conversion, the reads that originate from the acceptor site align onto the reference
sequence of the donor site which results in a twofold increase in sequencing depth relative to what is
expected (figure 1e-f).*

In addition to these textual changes we made a minor modification to figure 1e by adding coloured
bullets to the illustrated patients DNA. This to indicate that these nucleotides are present on both the
donor and acceptor side.

Furthermore, we explicity in the methods section that we have used bedtools for masking (modified text
in italics):

*Generating the masked reference genome*

When re-aligning the sequencing reads, all regions except 1 in a set of paralogs should be masked *in*
*the reference genome*. For this, we choose to mask the regions that had the least overlap with a protein
coding sequence (Supplementary data T252). *Masking was conducted with bedtools v2.28.0.*

I also find it strange that there is almost nothing written about the software package Chameleolyser,
what language is it written in, what is the input file format, how long does it take to run, what is the
output file format etc. There is a link to the GitHub page but I believe it will be of great importance for
people to know whether they have the computing infrastructure to run this tool, the correct file formats
and an idea of how long the analysis takes. If it takes a day to run a single sample then it will probably
not be applicable to other large cohorts without a serious commitment of resources.

We thank the reviewer for suggesting additional detailed information about the software implementation.
We addressed that by adding to the last paragraph in the introduction (modified text in italics):

Here, we present a method that enables the identification of SNVs, CNVs and ectopic gene conversions
in all paralogous regions in the coding portions of the human genome based on short-read sequencing
data, and apply it to a cohort of 41,755 WES samples. The method is called Chameleolyser *and is*
*implemented in Perl5. It requires a BAM or CRAM file (relative to GRCh37) as input and runs about one*
*hour on a single core for a single sample (depending on the enrichment kit and sequencing depth). Both*
*raw and filtered variants are written to a tab separated file. The tool is freely available on GitHub*
*(<https://github.com/Genome-Bioinformatics-RadboudUMC/Chameleolyser>) where also installation and*
*usage instructions can be found.*

And finally, although every new diagnosis is hugely important to that patient, only 25 new diagnoses
were made from 17,650 undiagnosed patients (0.14%). This shows that this method must be super
easy and quick to run and interpret if it is to be incorporated into routine genomic testing.

We agree with the reviewer that the increase in diagnostic yield is modest. Nevertheless, we believe
that it is relevant, especially for *STRC*, *OTOA* and *SMN1*. The running time is roughly 60 minutes for a
single sample on a single core, but parallelisation at the read re-alignment stage is possible.
Furthermore, for diagnostic settings, it is possible to only analyse known OMIM disease genes. In this
option the tool runs about 30 minutes on a single core for a single sample.

We would also like to emphasize the fact that for making a diagnosis we wanted to be 100% sure that
the genetic variant fully explains the phenotype and that the variant is truly present in the patients DNA.
For the latter we relied on orthogonal validation strategies which do not exist for most of the genes in
these difficult regions of the genome.

Furthermore, since we are to our knowledge the first group that systematically analysed the paralogous
regions of the genome, almost all rare variants are of unknown clinical significance. We are convinced
that when more research and diagnostic laboratories perform similar analysis and by the increased
number of long-read genomes that are becoming available, more and more of these variants will get a
clinical interpretation and the diagnostic yield increase by applying Chameleolyser will become higher.

Below is a list of questions/comments that highlight specific areas that I feel need to be explained in
more detail to make the manuscript more accessible and understandable:

1. Line 49 – You mentions CNV callers in the paragraph but highlight figure 1e which represents a SNV,
this is confusing.

We are happy to provide additional clarification in the text. We altered the 3rd and 4th paragraph of the
introduction as follows:

Despite their clinical relevance, gene conversions remain unidentified in the analysis of short-read data
such as whole-exome sequencing (WES) and whole-genome sequencing (WGS) data. Indeed, *in case*
*of an ectopic gene conversion, the sequencing reads that originate from the acceptor site will align onto*
*the reference sequence corresponding to the donor site. As a result, no reads will be aligned to the*
*acceptor site and short genetic variants (single or multi-nucleotide variants and small insertions and*
*deletions; jointly abbreviated to SNVs from here) that are introduced by means of the gene conversion*
*remain unidentified (Figure 1e). Copy number variant (CNV) callers however will typically identify such*
*events as deletions despite the fact that no deletion is present in the patients DNA (from here the term*
*‘deletion’ refers to genetic events with the size of a single or multiple exons).*

The issue of variant discovery within paralogous regions is not limited to gene conversions. *SNVs that*
*are not introduced by means of a gene conversion* also remain undetected, especially in *genomic*
*regions that have an identical paralog (100% sequence identity)*. In such cases, short sequencing reads
align equally well to multiple locations in the genome and will typically be assigned a mapping quality
of zero. These reads will be ignored by the variant calling algorithm as their alignment is deemed
ambiguous. As a result, *genetic* variants that are supported by these reads are not detected (Figure
1b).

2. Figure 1c, why is R2 not masked?

The identification of (homozygous) deletions and gene conversion events is based on the original
alignment (thus without any region being masked in the reference genome). We have generated and
used a masked alignment for the sole purpose of identifying SNVs which are not the result of a gene
conversion event. The clarifications in the text corresponding to the other comments should now also
make this more clear.

3. Line 57 – you mention methods to identify CNVs and say a limited number are specifically designed
to estimate copy number of paralogous regions but do not reference them, please add in refs.

The studies that we have found to describe copy number estimation of duplicated genomic regions
correspond to Handsaker *et al.*, 2015 and H.S.P. *et al.*, 2010 which were referenced 2 sentences further
in the text. We now referenced them right after the first sentence of the 5th paragraph in the introduction.

4. Can you explain why you only look at homozygous deletions or gene conversions, it's not immediately
clear to me why heterozygous events cannot be detected.

There are two technical challenges which make the identification of heterozygous events very
challenging or even impossible with short-read technologies. The first is the huge variability in read
depth (an intrinsic property of WES) which makes the distinction between 1 and 2 or 3 and 4 alleles
unclear. In case of a homozygous event, the difference with the reference sequence (homozygous) is

larger and thus, it is more reliable to discriminate these genotypes based on coverage. The second
challenge is the fact that ectopic gene conversions are typically small (order of magnitude of a single
exon) and it is much more difficult to identify such small events as opposed to larger events. Deletions
can definitely be larger, but there already exist decent tools for that purpose. We wanted to be highly
accurate for small events and therefore limited our call set to homozygous deletions and gene
conversions. In case ectopic gene conversions would be larger in general, we could gain accuracy and
the coverage-based estimate of the number of paralogous alleles could be done more accurate. We
did however study the heterozygous events pretty extensively but we were only able to identify these
events accurately for genes with a large number of deletions and conversion events in our dataset
combined with a relatively 'stable' coverage profile for the protein coding gene and its pseudogene (e.g.
*STRC*).

5. Results section, you give the total number of variants identified but could you give a per sample
overview.

To address this, we added 3 supplementary figures (Supplementary figure 1, 2 and 3) describing the
distribution of the number of variants per sample, respectively for SNVs which are not the result of a
gene conversion, deletions and gene conversion events.

6. Validation – what was the criteria for choosing the 20 samples that underwent long-read sequencing,
was it random or based on something else?

The 20 samples were selected for long-read sequencing because of the clinical interest of other
researchers (thus for the purpose of other research projects) which is independent of any variant or
region which we analyse in our work.

7. Identification of paralogous regions – can you reference where you obtained the lists of paralogous
regions.

We did not use any public set of paralogous regions. In the section 'Identification of paralogous regions'
in the methods we describe the method that we have used to generate our own list of paralogous
regions. For the regions in protein coding genes with known pseudogenes we have used Gencode 31
(lift 37). In the methods section of the manuscript we added the URL of this data file:

Starting from all pseudogenes in the comprehensive gene annotation file from Gencode 31 (lift37;
[https://ftp.ebi.ac.uk/pub/databases/gencode/Gencode_human/release_31/GRCh37_mapping/gencod
e.v31lift37.annotation.gff3.gz](https://ftp.ebi.ac.uk/pub/databases/gencode/Gencode_human/release_31/GRCh37_mapping/gencode.v31lift37.annotation.gff3.gz)), those with a corresponding protein coding gene were selected (n=1,680)

8. Line 278 – what does the software MAFFT do?

MAFFT is a bioinformatics tool that can be used for generating multiple sequence alignments for amino
acid or nucleotide sequences. In the methods we wrote, 'by using MAFFT v7.407²⁴, a multiple sequence
alignment was generated between each protein coding gene and its pseudogenes'.

9. Line 291 – what does EMBOSS Needle do?

EMBOSS Needle reads two input sequences and writes out their optimal global sequence alignment
by using the Needleman-Wunsch alignment algorithm. In the text we wrote 'To find the homologous
relations between the regions in the lists of regions we performed pairwise sequence alignments (PWA)
with EMBOSS Needle v 6.6.0.0²⁷ ...'.

10. Kernel density estimation and peak filtering – can you explain in more detail what KDE is and why
scikit-learn was used. Also reference them.

We now reference Pedregosa, Fabian, et al., 2011 for sci-kit learn in the kernel density paragraph in
the methods section. In addition to that, we now more carefully explain the idea behind the usage of
KDE. We have chosen scikit-learn because it is a very popular library to conduct machine learning
operations and it perfectly serves our goals. The complete section now becomes:

***Kernel density estimation***

*In order to accurately identify deletions and ectopic gene conversions it is important to consider a large*
*set of samples at once. Doing so, it can be seen from the coverage distribution if a poorly covered*
*sample is part of a wide distribution originating from samples without deletion or it is not part of such a*
*distribution and thus it is likely a sample with aberrant coverage due to a genetic event. The identification*
*of these coverage peaks is done in 2 steps. First we estimated the density of the data with the technique*
*of kernel density estimation (KDE). This technique smoothens the discrete data, i.e. it results in a*
*continuous curve with aligns with the density of the data. For this, scikit-learn was used (exponential*
*kernel; default parameters)³⁶. After having estimated the density we applied the argrelextrema function*
*from the scikit-learn software package to determine the local minima and local maxima of the curve.*
*This results in peaks or KDE clusters. This operation was done separately for deletions were both*
*paralogs in a set were deleted and deletions plus gene conversions affecting only 1 region in a set.*

11. In line 379 you use a frequency value of VAF and in line 398 you use percentage, please chose
one and use throughout.

We thank the reviewer for highlighting this discrepancy in our wording. We have chosen for a fraction
as it should be by definition (variant allele fraction) and we have corrected it throughout the text (also
within table 1).

12. Line 423 – change 'calls to were' to 'calls that were'.

We replaced 'to' to 'that' in the revised text.

13. Line 425 – what is meant by the term 'automated fashion'?

We meant that we first programmatically compared the variants from our analysis with the VCFs from
the long-read data that were produced by DeepVariant. Next, we visually checked the remaining
variants in IGV, which is a more manual approach. We agree that it is not relevant for the validation
which part of the validation is automated and which part is not. For that reason we replaced 'in the
automated fashion' by 'using the VCFs'.

Reviewer #2 (Remarks to the Author)

Major: the authors do not benchmark Chameleolyser against other CNV and variant calling methods.
In duplicated exons, Chameleolyser is able to detect (i) large deletions, (ii) sequence variants, (iii) gene
conversion events. Even though copy number variation detection using WES is not as advanced as
with WGS, there is a number of CNV-detection tools that use WES data (Zhao et al., BMC Bioinf.,
2020). The authors should compare against these methods to show that Chameleolyser performs better
than existing tools.

We thank the reviewer for his/her comment. We have now run ExomeDepth and Conifer on our study
cohort and describe the comparison with Chameleolyser in a new subsection ('Comparison with other
variant callers') in the results section of the manuscript.

*Chameleolyser's ability to identify SNVs (not the result of a gene conversion) was compared with GATK
and DeepVariant¹⁶. The sensitivity for both of these tools is exactly zero within genomic regions that
are associated with zero mapping qualities in WES (Supplementary data T9, Figure 1b), With
Chameleolyser a sensitivity of 43% is achieved (Methods). In regions onto which sequencing reads
align uniquely, it has been shown that GATK and DeepVariant are excellent tools for the identification
of SNVs¹⁷. Within these regions, the added value of Chameleolyser is limited as it increases GATK's
sensitivity from 86.3% to 88.0% (Methods).*

*Sensitivity could not be assessed for homozygous deletions and ectopic gene conversions since we
cannot, due to the availability of only a limited number of long-read sequencing samples, derive a call
set of high-quality events with a population allele frequency ≤ 0.10 (Methods). The unique value of
Chameleolyser can however be demonstrated by comparing its output with ExomeDepth¹⁸ and
Conifer¹⁹ (Methods, Supplementary figure 4). Within the 20 in-house samples for which LRS alignments
were generated, there are 4 events (3 deletions and 1 gene conversion) that are only called by
Chameleolyser. Of these, 2 events (1 deletion and 1 conversion) were concordant with the LRS
alignments. The other 12 deletions and 7 conversions that were identified by Chameleolyser (all
concordant with LRS) are all called as deletions by ExomeDepth. As opposed to Conifer (for which
there are no homozygous deletion calls within the validation samples), ExomeDepth made an additional
201 homozygous deletion calls which were not made by the other tools. Based on the LRS alignments
we estimated the precision at 32.5% (Methods, Supplementary data T10).*

The authors should also compare their variant calling performance against other short variant callers,
for example GATK, Freebayes and DeepVariant. Even if these tools are not designed for duplicated
regions, the authors need to prove that their method works better than existing variant callers.

We agree with the reviewer that a comparison with other variant callers was missing. For that reason
we have run GATK (implicit in the method) and DeepVariant on 25 samples (20 in-house + 5 genome-
in-a-bottle samples) for which we both had WES and long-read genome sequencing data. From this we
could conclude that Chameleolyser had a sensitivity of 43% while for GATK and DeepVariant this is
exactly 0 within genomic regions that have an identical copy elsewhere in the genome. We describe
these results within the new subsection 'Comparison with other variant callers' within the results.

*Chameleolysers ability to identify SNVs (not the result of a gene conversion) was compared with GATK*
*and DeepVariant¹⁶. The sensitivity for both of these tools is exactly zero within genomic regions that*
*are associated with zero mapping qualities in WES (Supplementary data T9, Figure 1b), With*
*Chameleolyser a sensitivity of 43% is achieved (Methods). In regions onto which sequencing reads*
*align uniquely, it has been shown that GATK and DeepVariant are excellent tools for the identification*
*of SNVs¹⁷. Within these regions, the added value of Chameleolyser is limited as it increases GATK's*
*sensitivity from 86.3% to 88.0% (Methods).*

Major: Homozygous SNVs can be identified using standard variant calling (as they would appear on all
repeat copies irrespective of read mapping). The authors should show that their method better identifies
homozygous variants than standard variant callers.

We thank the reviewer for this comment. We however disagree that homozygous SNVs in paralogous
regions can be identified with standard variant calling methods (as shown above). The reason for this
is that the reads that align onto these paralogous regions have zero mapping qualities. Irrespective of
the zygosity of a variant, the supporting reads are not considered by the variant calling algorithms
because of that. Furthermore, the comparison of variant callers in the comment above applies to both
heterozygous and homozygous events.

Major: 8/8 gene conversion events and 11/15 deletion events were validated (line 99). The authors write
that the four remaining deletion events could not be validated as LRS alignments were unclear.
However, the validation was performed manually (line 422). Consequently, the authors should describe,
what are their criteria for low LRS clarity, and how many of the remaining 8 gene conversions and 11
deletions have low LRS clarity.

In case of a deletion it is possible that some LRS reads partly align onto the region which is actually
deleted. This happens because the reads are not necessarily clipped at the breakpoints of the deletion
(in case the length of the deletion is of the size of the LRS read or longer). These read endings that
align within the deleted region are then full of non-matching bases. Previously we defined these LRS
alignments to be unclear. In our revised work we have improved the validation by blasting these read
tails and verifying whether these read portions indeed originate from the region up or downstream (they
should correspond without non-matching bases). Doing so, we could conclude that 13/15 deletion calls
correspond to the LRS data. We have described the validation procedure in the methods section.

*All 15 deletion calls were visually inspected in the LRS alignments. For 8/15 deletion calls there is a*
*maximum of 1 LRS read aligning onto the region that Chameleolyser claims to be deleted (as opposed*
*to samples without the deletion). For 5 of the remaining 7 deletions the reads that align onto the region*
*corresponding to the deletion call are all read tails with a large number of non-matching bases. We have*
*blasted these read tails and could conclude that these read portions actually correspond to the region*
*up or downstream of the actual deletion (Supplementary figure 11). The 2 remaining deletion calls are*
*in extremely complex regions (the KIR cluster) and we could not unequivocally come to the same*
*conclusion. By comparing the alignments with samples without the deletion call we can definitely see*
*that there is a genetic event, but not necessarily a homozygous deletion. For that reason we conclude*
*to have 13/15 deletion calls which correspond to LRS alignments.*

*All conversion calls were visually inspected in the LRS alignments. Here, there should not be a deletion*
*or coverage drop. There should be homozygous variants which correspond to the sequence differences*
*between the linked donor and acceptor site. We could confirm this for all 8 ectopic gene conversion*
*calls.*

It is not required, but the paper would benefit from higher sample size. In order to achieve that, the
authors may analyze samples with publicly available WGS/WES data and PacBio HiFi data, such as a
subset of samples from the 1000 genomes project. Alternatively, the authors can analyze 7 samples,
collected by the Genome in a Bottle consortium (GIAB), that have high-confidence variant call sets,
which can be used in an additional validation step.

We have run Chameleolyser on 5 GIAB samples, *i.e.* the samples for which both WES and PacBio LRS
exist. We added the results of this analysis to the validation section within the results of the manuscript.

*In addition to our in-house validation samples we also applied Chameleolyser on 5 genome-in-a-bottle*
*samples (Methods). Since the identification of deletions and gene conversions requires a larger number*
*of samples enriched with the same enrichment kit, the precision analysis was restricted to SNVs (not*
*the result of a gene conversion). From the 118 SNV calls made by Chameleolyser, 98 are concordant*
*with LRS (83.1%; Methods, Supplementary data T8). From the 39 calls corresponding to rare SNVs,*
*35 were concordant with LRS (89.7%; Supplementary data T8).*

Major: Authors rely on singly unique nucleotides (SUNs, also known as paralogous-sequence variants,
or PSVs) to identify gene conversion events. However, it has been shown before (Sudmant et al.,
Science, 2010; Mueller et al., Am.J.Hum.Genet., 2013), that many SUNs are polymorphic in the
population. In this case, relying on such SUNs can lead to incorrectly identified gene conversion events.

We have read both papers with great interest and agree with the reviewer that many SUNs are
polymorphic in the population. However, this does not lead to incorrectly identified gene conversions.
We would like to explain this by using an example. Suppose there is a protein coding gene with an A-
allele in the reference sequence and a pseudogene with a C-allele on the corresponding position. We
assume all other nucleotides to be identical. The method Chameleolyser uses the number of reads that
uniquely align onto a certain region to quantify the number of A and C alleles. Suppose in a certain
individual there is a homozygous variant from A to C, either rare or common in the population, then
Chameleolyser will identify 4 C-alleles. In this scenario we will assume that ectopic gene conversions
(in some generation above the individual at study) have caused the introduction of the C-alleles in the
protein coding gene. Theoretically, it could also be that *de novo* SNVs which are not the result of ectopic
gene conversions have caused the A to C change in the protein coding gene, but we believe that this
is by far less likely. In addition, the genetic result is indistinguishable with the gene conversion scenario,
*i.e.* a protein coding gene with 2 C-alleles, and this is what we want to identify.

Major: The methods are ad-hoc, with many arbitrarily selected thresholds and parameters (for example,
lines 287, 295, 341-2, 349, 351-5, 361-3, 378-9, 394, 398-9, 401, 405, 438). The authors should aim to
reduce the number of parameters and explain how parameter values were selected where possible.

We agree with the reviewer that using fewer parameters would contribute to an easier to follow methods
section. Although we are not able to reduce the number of parameters we have added additional
explanations showing that the parameter values were not chosen arbitrarily. Several of these parameter
values have a biological (e.g. 90% sequence identity between linked donor and acceptor regions) or
statistical (e.g. the threshold for variant allele fraction (0.15)) origin, others are routinely used in literature
(e.g. thresholds for GATK quality scores). We have referenced the relevant papers where necessary in
the methods section. There are also some parameters (values) that are very specific to this work. A full
parameterisation of these parameters would not be realistic because of the computational cost and
because of the lack of a true positive dataset (there isn't a set of true positive ectopic gene conversions).
Since we are the first group to perform such an analysis we aimed for highly accurate results which are
relevant for rare disease patients. Nevertheless we explained and motivated in the methods section the
choices we made for each of these parameter values.

Minor: The method identifies disease-causing mutations in just a handful of well-known genes (STRC,
OTOA, SMN1). Is the method applicable in other disease-relevant duplicated genes?

The method indeed works and is applied on all paralogous regions within the exome. It is however not
trivial to validate variants in complex regions so we focused on genes for which we have multiple assays
(e.g. MLPA and long-range PCR primers). In addition to that, since these regions are largely
unexplored, most of the rare variants within these regions are yet of unknown clinical significance.

Minor: Are there any estimates of the method sensitivity, i.e. the percentage of variants that are present
in the data and found/not found by Chameleolyser? If the method has low sensitivity, explain why it is
not important in this case.

For SNVs (not the result of a gene conversion and within regions that are associated with zero mapping
qualities) we estimated the sensitivity to be 43% whereas for GATK and DeepVariant this is exactly 0.
We have written these results in the paragraph 'Comparison with other variant callers' within the results.
This sensitivity might seem low but it clearly is substantially different from existing variant callers.
Furthermore, for genetic testing it is much more important to have a high precision (> 80%).

Minor: The authors seemingly use the term "SNV" for both short mismatches and indels, and use term
"deletion" for gene/exon deletion events. In my opinion, this is a bit unclear. Perhaps, the authors can
use terms "sequence variants", "short variants", "SNVs and indels", etc. to better underline the
difference between two concepts.

We agree with the reviewer that it is essential to clearly discriminate between these different types of
genetic variants since the underlying methods to identify them are very different. For that reason we
have abbreviated all short variants to SNVs and we did that systematically throughout the text. This
category thus includes single and multi-nucleotide variants as well as small insertions and deletions.
The sizes of the insertions and deletions correspond to the capabilities of the short variant caller in
short-read next-generation sequencing data. Deletions that go beyond the detectability of short-variant
callers in short-read data are systematically called 'deletions' throughout the text. We defined these

different types of genetic events in the third paragraph of the introduction and from there onwards
systematically use these conventions.

Despite their clinical relevance, gene conversions remain unidentified in the analysis of short-read data
such as whole-exome sequencing (WES) and whole-genome sequencing (WGS) data. Indeed, *in case*
*of an ectopic gene conversion, the sequencing reads that originate from the acceptor site will align onto*
*the reference sequence corresponding to the donor site. As a result, no reads will be mapped to the*
*acceptor site and the short genetic variants (single or multi-nucleotide variants and small insertions and*
*deletions; jointly abbreviated to SNVs from here) that are introduced by means of the gene conversion*
*remain unidentified (Figure 1e). Copy number variant (CNV) callers however will typically identify such*
*events as deletions despite the fact that no deletion is present in the patients DNA (from here the term*
*'deletion' refers to genetic events with the size of a single or multiple).*

Minor: Line 85: 2,191,707 SNVs. Does this number include the same heterozygous variant split into
several positions? If yes, the number is misleading. Same on line 146.

No, 2,191,707 is the number of variant calls which is the same as the number of variants (apart from
the very rare situations where there is a homozygous variant in all paralogs). Once annotated these
variants need to be "split" and we talk about VAPs.

Line 388: True number of variants lies between 56M and 119M. Saying that there are 119M variants is
misleading. Same goes to lines 394-407, perhaps some other terminology or abbreviation can be used.

We agree with the reviewer that using the word 'variants' here is incorrect. We replaced that with 'VAP',
a term that we specifically invented for this duality. The same holds true for the numbers in the filter
process.

Minor: Line 108-...: explicitly mention if SUNs are discarded, or considered as VAPs?

Variant calls originating from SUNs and thus also variants corresponding to SUNs are not considered
in the analysis of SNVs (not the result of gene conversions). This is the first step after variant calling
with LoFreq on the masked alignments (cf. methods). We now also clarified this in the beginning of the
paragraph.

Heterozygous SNV calls (not due to a gene conversion *and not corresponding to SUNs (methods)*)
result from a genomic alteration in one of the paralogs within the respective set of paralogs (Figure 1b).

Minor: Line 125: better to use terminology "p-value" than simply "p".

We replaced p by p-value throughout the text.

Minor: Line 128: mention, that in non-duplicated (unique) regions. Also, the next sentence may be a bit
easier to read, if it would be fractions/percentages instead of amounts relative to synonymous variants.

We have replaced 'non paralogous' to 'non-duplicated (unique)'. In the next sentence we express the
number of missense variants relative to the number of synonymous variants. This ratio is what we need
in the next step. The percentage of variants that has a missense consequence is another number which
we don't really need for the point we are making in that paragraph. So, it is unclear to us how we could

use percentages or fractions to express the value we want to express (a kind of synonymous-
normalized number of missense and LoF variants).

Minor: Line 140: “We filtered for” may be better to replace with “We selected” or other phrase, otherwise
it is ambiguous (can be “we discarded”).

We have adopted the suggestion of the reviewer.

Minor: Line 294: What happens if a paralog has similarity > 0.9 with some paralogs in the set, but < 0.9
with other paralogs? Line 296: what about regions that have similarity > 0.9 with two sets?

Within a set of paralogs, all sequences have a similarity score ≥ 0.9 with each other. None of these
sequences is allowed to have a similarity score ≥ 0.9 with a sequence in another set of paralogs. It is
indeed possible that a sequence has a similarity < 0.9 with a paralog in another set of paralogs. In order
to conduct the analysis we aimed for we had to discretize a continuous property which is indeed an
approximation/simplification of evolutionary reality.

Minor: Line 319: Provide more description about GATK and LoFreq variant calling. Do the authors use
GATK on the input WES data, and LoFreq on the reads, re-mapped to the masked genome?

Yes that is true. The reason to use GATK on the input WES data is to subtract all of these variants from
our analysis. In the methods section we now explicitly state that LoFreq was used on the ‘masked
alignment’:

Therefore we used LoFreq 2.1.3.1 for sensitive variant calling *on this newly generated alignment*
(parameters: no-default-filter, use-orphan, no-baq, no-mq, sig=1).

In addition to that we also added explicitly that we use the GATK calls from the original alignment in the
variant filtering to subtract from the call set. The fourth criterium now is: The variant is not present in the
original VCF (*GATK variant calling; no masking*)

Minor: Line 329: Why only 2-copy regions are analyzed? Does the method work in higher-copy
duplications, for example NCF1, SRGAP2, etc.?

Since the vast majority of paralogous sets consists of exactly 2 members and since there is a substantial
drop in accuracy for higher-copy numbers, we restricted this part of the analysis to sets of paralogs with
2 members. The fact that most of the paralogous sets indeed have 2 members can be derived from
Supplementary data T20 and Supplementary data T23. In total we have 13,672 regions of which 9,844
are part of a set consisting of 2 members (a very similar conclusion can be read in Ebbert et al., 2019).
For sets consisting of 3 members or more it is very challenging (if possible at all) to identify homozygous
deletions or ectopic gene conversions of 1 of the members. Here we want to draw attention to the fact
that we have designed this analysis for whole exome sequencing data which is much more variable in
coverage as compared to whole genome sequencing data. The tools that were published previously
and that are able to estimate a total copy number for sets of paralogs (which is different from the aim
of our method) are designed for WGS only where much more data points are available and a more
uniform coverage is achieved.

Minor: Line 336: Need more details about KDE, what parameters are used, if there is any post-
processing.

In the paragraph 'Kernel density estimation' in the method section we added that we have used default
parameters for KDE. In that paragraph we also explained the post-processing we did.

*In order to accurately identify deletions and ectopic gene conversions it is important to consider a large*
*set of samples at once. Doing so, it can be seen from the coverage distribution if a poorly covered*
*sample is part of a wide distribution originating from samples without deletion or it is not part of such a*
*distribution and thus it is likely a sample with aberrant coverage due to a genetic event. The identification*
*of these coverage peaks is done in 2 steps. First we estimated the density of the data with the technique*
*of kernel density estimation (KDE). This technique smoothens the discrete data, i.e. it results in a*
*continuous curve which aligns with the density of the data. For this scikit-learn was used (exponential*
*kernel; default parameters)³⁶. After having estimated the density we applied the argrelextrema function*
*from the scikit-learn software package to determine the local minima and local maxima of the curve.*
*This results in peaks or KDE clusters. This operation was separately done for deletions were both*
*paralogs in a set were deleted and deletions plus gene conversions affecting only 1 region in a set.*

Minor: Line 370: Specify merge distance or criteria.

The criterium is 3 intermediate genes which was written in the text. We now have clarified that as
follows:

Next, deletions and conversions in neighbouring genes were merged. *Also when 1, 2 or 3 coding genes*
*(but not necessarily part of a paralogous set of sequences) exist between the 2 different CNV calls, it*
*was assumed that these originate from the same genetic event and therefore these calls were merged*
*into 1 call.*

Minor: In the results (line 105), the authors say that the 99% of SNVs present in the offspring are called
in one of the parents. Does this mean that in methods (line 430) that the variant in a child and in a
parent have VAF $\geq 15\%$? Does this include very common variants? If yes, the authors can show the
numbers for rare variants as well.

With respect to possible de novo variation we can indeed discriminate 3 groups of variants.

- 1) VAF child ≥ 0.15 and VAF of both parents < 0.01 . These are considered de novo
- 2) VAF child ≥ 0.15 and VAF of 1 or 2 parents ≥ 0.15 . This is considered inherited.
- 3) VAF child ≥ 0.15 and VAF of both parents between 0.01 and 0.15. These variants are also
considered as inherited.

The variants in the third category are definitely called in the parents but they will not end up in our
2,191,707 variant calls which we had after filtering. The de novo analysis was conducted for a cohort
allele frequency threshold of 0.10 and 0.005 (Supplementary data T23) with similar results for both of
the thresholds. The major purpose of this analysis is to show that our variant filtering works, i.e. we
don't have a large number of false positives.

Reviewer #3 (Remarks to the Author)

Minor:

1. Clarify that unique sequence to identify ectopic events is generally only necessary for short-read
sequencing technologies because long-read technologies will generally have reads long enough to
anchor within adjacent unique sequence.

a. Suggested edit: "In order for ectopic gene conversion event to be detectable using short-read
sequencing technologies, the donor and acceptor site should differ in at least one nucleotide."

We thank the reviewer for his comment but we don't entirely agree. In case the donor and acceptor
sequence are 100% identical it will always be unknown whether or not an ectopic gene conversion
event has happened. This also holds true for long-read sequencing because the specific region that is
part of the conversion event is identical and thus indistinguishable from the original sequence. However,
given comment number 3 we believe that the complete sentence can be removed as we just wanted to
explain that new genetic variation might be introduced in the acceptor site, variation that might lead to
disease. The issue of detectability is discussed later in the manuscript.

2. Correct definition of ectopic gene conversion. Authors state that ectopic conversion events are copied
from one genomic region to "a distant homologous region" (i.e., the acceptor site). The acceptor site
itself *could* be homologous, but does not have to be, and often is not.

a. Suggested edit: Remove the word "homologous".

We thank the reviewer for this comment. We took over his suggestion by removing the word
'homologous'.

3. Transition between sentences doesn't quite follow, logically. In one sentence, the authors state that
the two paralogous regions must differ by at least one nucleotide to be *detected* (for short-read
technologies) and the following sentence uses this as justification that these events introduce new
genetic variation. It's true that these events introduce genetic variation, but the logic is a bit
disconnected.

We agree with the reviewer that the paragraph was not completely clear. To bring more clarity we
removed the condition for gene conversion detectability (at least 1 nucleotide which is different between
donor and acceptor). We instead wrote '*When the donor and acceptor sequence differ, new genetic
variation is introduced into the acceptor site*'.

4. Figure 1e. Maybe I'm missing something, but I do not understand why the authors state that reads
from R1 (what I believe is the acceptor site) would only align to R2 (what I believe is the donor site).
Specifically, I do not understand what the authors are saying in the third paragraph of the introduction.
In paragraph two, the authors are talking about the paralogous regions as though they have unique
sequence. If they have sufficiently unique sequence, which Figure 1e suggests, then why would reads
from R1 only align to the donor site (R2)? If the R1 is sufficiently unique and the paralogous genes are
included in the reference genome, then the aligner would certainly align R1 reads to R1. If R1 and R2

are *not* sufficiently unique, then the aligner would assign the reads a mapping quality of zero (0) and
randomly assign the reads to both R1 and R2. Figure 1e clearly shows reads from R2 aligning to R2,
but there are zero R1 reads present anywhere. i.e., R1 reads are not aligning to R2 in Figure 1e as
paragraph 3 suggests they would (though I don't see why they would).

We agree with the reviewer that if R1 and R2 are sufficiently unique the sequencing reads from R1 will
align onto the reference sequence corresponding to R1 and reads originating from R2 will align onto
R2. This is actually illustrated in figure 1d (the 'no variant' scenario). Figure 1e represents the scenario
in which there is an ectopic gene conversion from R2 to R1. Because of the short sequencing reads,
the reads that originate from R1 will align to R2. In the patient's DNA R1 and R2 are identical but in the
reference genome R1 and R2 are different. So all of the reads that originate from R1 and from R2 in
the patient's DNA will align onto R2 in the reference genome. Therefore we added more reads to the
illustration (aligning to R2 in fig 1e). Since the sequences are identical (the patient's R1 and R2) they
appear as they originate from R2 but this is not the case. For that reason we can only resort to coverage
analysis to derive whether or not an ectopic gene conversion event is present or not. We clarified this
in the legend of figure 1:

Chameleolyser also considers the coverage at locus R2. As a result, gene conversions can be
distinguished from deletions. *Indeed, in case of an ectopic gene conversion, reads that originate from*
*the acceptor site will align onto the reference sequence of the donor site resulting in a two-fold increase*
*of the sequencing coverage as opposed to the scenario where no gene conversion is present.*

5. Because the nuances of this work are so critical, the authors need to explicitly state when they're
talking about variants in the patient's DNA vs differences between genomic regions. e.g., I'm intimately
familiar with these nuances, but even I had to re-read paragraph 4 of the introduction to understand
what the authors were referring to.

Thank you for highlighting the challenges in paragraph 4. We slightly modified the paragraph to make
it more clear.

The issue of variant discovery within paralogous regions is not limited to gene conversions. *SNVs that*
*are not introduced by means of a gene conversion* also remain undetected, especially in *genomic*
*regions that have an identical paralog (100% sequence identity)*. In such cases, short sequencing reads
align equally well to multiple locations in the genome and will typically be assigned a mapping quality
of zero. These reads will be ignored by the variant calling algorithm as their alignment is deemed
ambiguous. As a result, *genetic* variants that are supported by these reads are not detected (Figure
1b).

6. At risk of being self-serving, while many aspects of the authors' algorithm are unique, they also use
many of the same methods we described in our 2019 paper but did not cite our paper. For example,
the authors specifically mention Ebbert et al. 2019 in the introduction, but do not explicitly cite the paper.
Additionally, they do not give credit in the methods or results where they probably should.

We apologise that we forgot to reference your work. Right after the sentence starting with 'Ebbert et al.,
2019' we now explicitly referred to your work. We also cited your work after the first sentence of the
results section as this describes the masking part of the method.

7. Figures are low quality in the draft I received.

We agree that clear figures are essential for understanding our work. We have all figures in svg and
high resolution png (1200 dpi), but probably by pasting these images in the MS Word the quality
dropped.

8. "patients dna" in Figure 1 should be "Patient's DNA"

We have changed 'patients dna' into 'patient's DNA'.

9. Many of the terms the authors use are ambiguous. e.g., the title for table 2 is unclear whether the
authors are claiming they discovered a new pathogenic variant that has never been associated with
disease, or that the variants themselves were known to be pathogenic, but hadn't been identified in the
patients.

We have changed the title of Table 2 to 'Overview of genetic diagnosis in our study cohort as a
consequence of disease-causing variants identified with Chameleolyser'.

10. Table 2 is also missing critical information on the variants discovered. Authors should include the
chromosomal position. In the case of a deletion, they should specify exactly what is deleted.

This comment is not completely clear to us. The legend states 'The second, third and fourth column
respectively represent the chromosome, genomic start and end of the event (hg19)' which is indeed
reflected in the table.

11. IGV screenshots showing examples of some of these events would be very helpful for readers.

We agree with the reviewer that adding IGV screenshots to the manuscript will help the reader to
capture the main message. For that reason we added 3 supplementary figures (supplementary figure
7, 8 and 9) in which examples of pathogenic variants in *STRC*, *OTOA* and *SMN1* can be found. For
each of these genes, an example of each type of variant is displayed.

12. In some areas of the Results, it's difficult to know whether the authors are claiming new variant
associations, or identifying known pathogenic variants in undiagnosed patients. The authors should be
super clear about this in every instance.

To avoid this confusion, we removed the word 'novel' where it could lead to confusion.

13. The authors should distinguish between evolutionary gene conversion events and de novo events,
where appropriate.

The identification of heterozygous ectopic gene conversion events is nearly impossible for most of the
genes in our analysis (technical reasons). For that reason we restricted our coverage dependent
analysis to homozygous events. As a result, it is unlikely that the actual gene conversion arose in the
individual that we measure. It is much more likely that all of the ectopic gene conversions that we identify

are alleles (pathogenic or not) that circulate in the general population. The wording 'identify an event'
could indeed, to some extent, suggest a *de novo* characteristic. We clarified this in the text where
appropriate. In the last paragraph of the introduction we replaced 'identification of ectopic gene
conversion events' to 'identification of ectopic gene conversions' and in the results section where we
first report our ectopic gene conversion findings we added a part about their origin as follows (new text
in italics): All of the homozygous variants are introduced in the gene of interest by means of gene
conversions *that most likely occurred in a proximal or distant ancestor* (a total of 21).

Reviewers' Comments:

Reviewer #1:

Remarks to the Author:

I would like to thank the authors for responding so thoroughly to the questions/comments I made following the first review. Having read the rebuttal document I am satisfied that the resubmitted manuscript has been edited to take into consideration all the points I raised and feel it is sufficiently complete for publication.

This an interesting method which has the potential to identify pathogenic variants that would otherwise be missed using routine clinical testing. It should therefore be of interest to a wide-range of clinical diagnostic labs globally.

Best wishes

Hywel J Williams (Cardiff University)

Reviewer #2:

Remarks to the Author:

The authors have addressed all of my concerns.

Reviewer #3:

Remarks to the Author:

I think the edits are satisfactory